# Basin inversion: Reactivated rift structures in the central Ligurian Sea revealed by OBS

Martin Thorwart[1], Anke Dannowski[2], Ingo Grevemeyer[2], Dietrich Lange[2], Heidrun Kopp[1,2], Florian Petersen[2], Wayne C Crawford[3], Anne Paul[4] and the AlpArray Working Group[5]

[1] CAU, Institute of Geosciences, Christian-Albrechts-Universität zu Kiel, 24105 Kiel, Germany

[2] GEOMAR, Marine Geodynamics, Helmholtz Centre for Ocean Research Kiel, 24148 Kiel, Germany

[3] IPGP, Laboratoire de Géosciences Marines, Institut de Physique du Globe de Paris, Paris, 75238 Cedex 5, France

[4] Univ. Grenoble Alpes, Univ. Savoie Mont Blanc, CNRS, IRD, UGE, ISTerre, 38000 Grenoble, France

[5] For the complete team list visit the link which appears at the end of the paper.

*Correspondence to*: Anke Dannowski (adannowski@geomar.de) or Martin Thorwart (martin.thorwart@ifg.uni-kiel.de)

**Abstract.** The northern margin of the Ligurian Basin shows notable seismicity at the Alpine front, including frequent magnitude 4 events. Seismicity decreases offshore towards the Basin centre and Corsica, revealing a diffuse distribution of low magnitude earthquakes. We analyse data of the amphibious AlpArray seismic network with focus on the offshore component, the AlpArray OBS network, consisting of 24 broadband ocean bottom seismometers deployed for eight months, to reveal the seismicity and depth distribution of micro-earthquakes beneath the Ligurian Sea.

Two clusters occurred between ~10 km to ~16 km depth below sea surface, within the lower crust and uppermost mantle. Thrust faulting focal mechanisms indicate compression and an inversion of the Ligurian Basin, which is an abandoned Oligocene-Miocene rift basin. The basin inversion is suggested to be related to the Africa-Europe plate convergence. The locations and focal mechanisms of seismicity suggest reactivation of pre-existing rift-related structures. Slightly different striking directions of presumed rift-related faults in the basin centre compared to faults further east and hence away from the rift basin may reflect the counter-clockwise rotation of the Corsica-Sardinia block. High mantle S-wave velocities and a low Vp/Vs ratio support the hypothesis of strengthening of crust and uppermost mantle during the Oligocene-Miocene rifting-related extension and thinning of continental crust.

## 1 Introduction

Earthquakes of magnitude 4 are frequently recorded in the Ligurian Basin, which is a plate interior. Especially the Ligurian margin at the junction between the southwestern Alps and the Ligurian Basin (hereafter named the Alps-Liguria junction) is active with maximum magnitudes of Mw 6 to 6.5, indicating a moderate seismic activity with occasionally strong earthquakes (Béthoux et al., 1992; Courboulex et al., 2007; Béthoux et al., 2008; Larroque et al., 2012, 2016; Manchuel et al., 2017). Seismic activity is highest along the Côte d'Azur and the Ligurian coast and decreases towards the central basin and Corsica

(Fig. 1). The Ligurian Sea formed during Oligo-Miocene times as a back-arc basin (Burrus, 1984; Rehault et al., 1984; Faccenna et al., 1997; Gueguen et al., 1998; Rosenbaum et al., 2002), but extension stopped ~16 Ma. Nocquet and Calais (2004) have shown that the most of the plate convergence between Africa and Europe is accommodated at the Maghrebian chain. A present day horizontal convergent motion of 0.4 mm/year is observe between the Corsica-Sardinia block and mainland Europe (Nocquet, 2012; Masson et al., 2019). Compressive earthquakes occasionally occur in north dipping reverse faults at the northern margin, indicating basin inversion (Larroque et al., 2011; Sage et al., 2011). The interior of plates is, in general, predominantly aseismic (McKenzie and Parker, 1967). In these relatively stable tectonic regions, sparse seismicity may represent diffuse deformation and is commonly related to the reactivation of pre-existing fault planes (Zoback, 1992). In the Tyrrhenian Sea, the Africa-Europe convergence caused reactivation of pre-existing fault planes, revealing an inversion of the basin that is mainly observed along the margins (Zitellini et al., 2020).

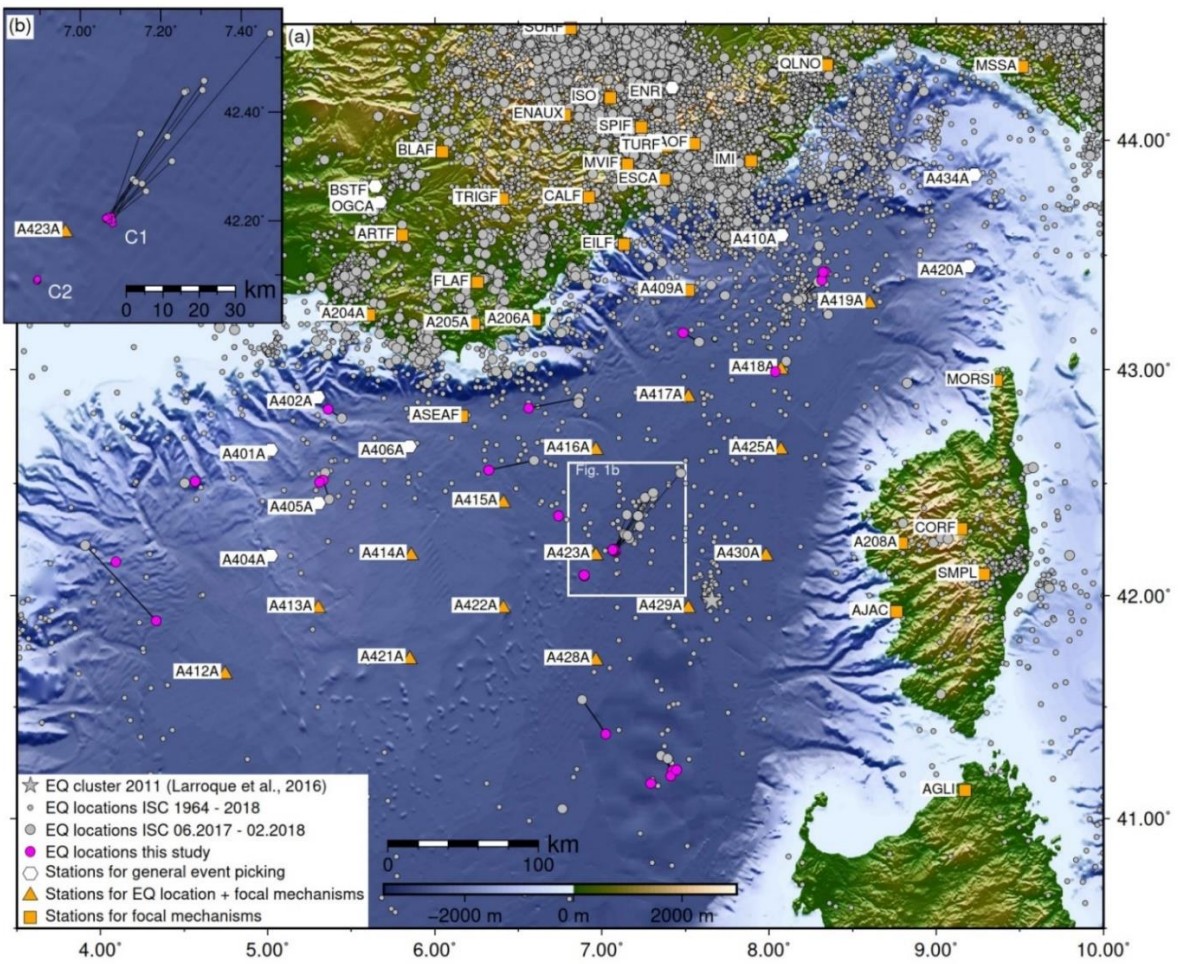

**Figure 1: Topographic map of (a) the Ligurian Sea and (b) the area where the cluster events occurred (GMRT data, Ryan et al., 2009). Grey circles mark the epicentres of earthquakes from the ISC bulletin (Storchak et al., 2017). Magenta circles mark the epicentres of earthquakes within the Ligurian Basin observed in this study. Black lines connect locations of the same events found in both the ISC bulletin and this study.**

Here, we report on local seismicity in the centre of the Ligurian Basin (Fig. 1a). We analysed two earthquake clusters (Fig. 1b) that were recorded by an amphibious seismic network operated in the framework of the European AlpArray initiative (Hetényi et al., 2018). The AlpArray ocean bottom seismometer (OBS) network, a long-term broadband OBS array, recorded ground motion continuously for eight months (06.2017-02.2018). The long-term OBS deployment enables robust source estimates of earthquakes far away from land stations. Nevertheless, observations from the land stations improved the estimate of fault plane solutions. The observed earthquake clusters and their depth distribution provide constraints on the crust and upper mantle rheology and provide insights into the current regional stress field.

## 2 Geological and geodynamic setting

The Western Mediterranean, including the waters west of Apennines and Sicily, consists of several basins (Fig. 2) that formed from Oligo-Miocene times to the present (e.g. Burrus, 1984; Rehault et al., 1984). The geodynamic setting was controlled by the convergence of the African and Eurasian plates (e.g. Dercourt et al., 1986; Nocquet and Calais, 2004; Serpelloni et al., 2007) and the rollback of the Apennines, Calabrian and Gibraltar subduction zones (Jolivet and Faccenna, 2000). The fast rollback of the westward migrating Gibraltar arc and the south-eastward retreating Apennines-Calabrian arc played a major role in the opening of the Mediterranean sub-basins (Frizon de Lamotte et al., 2000; Mauffret et al., 2004; Handy et al., 2010). The Ligurian Basin, opening ~30-16 Ma (Burrus, 1984; Bache et al., 2010), and the Alboran Basin, opening ~27-8 Ma (Comas et al., 1999), are the oldest basins in the Western Mediterranean. The Algerian Basin, opening ~16-8 Ma (Mauffret et al., 2004), and the Tyrrhenian Basin, opening ~8-0 Ma (Faccenna et al., 2001, 2004; Rosenbaum et al., 2002), are the youngest basins in the Western Mediterranean. In the latter two basins, geophysical and geological data clearly show that extension caused break-up and seafloor spreading (Nicolosi et al., 2006; Bouyahiaoui et al., 2015; Prada et al., 2016; Booth-Rea et al., 2018), while the Alboran Basin is a domain of extended continental crust modulated by arc magmatism (Booth-Rea et al., 2018; Gómez de la Peña et al., 2020). The SW part of the Liguro-Provencal Basin is floored by 'atypical' oceanic crust (Gailler et al., 2009; Afilhado et al., 2015; Moulin et al., 2015). However, the extent and nature of the oceanic domain towards the northeast into the Liguro-Provencal Basin remains debated.

The Ligurian Basin underwent a long-lasting phase of extension from Late Oligocene to Miocene (Rehault et al., 1984; Gueguen et al., 1998; Finetti et al., 2005), progressively opening from south to north with a high synrift sedimentation (Sage et al. (2011). Based on magnetic and seismic data, oceanic spreading with unroofing of mantle material was proposed for the late opening period ~21-16 Ma (Le Douaran et al., 1984; Pascal et al., 1993; Contrucci et al., 2001; Rollet et al., 2002; Speranza et al., 2002). For the Ligurian Basin, which is the NE part of the Liguro-Provencal Basin, a recent analysis of a seismic refraction profile proposes that rifting failed before oceanic spreading initiated, while it remains unclear if mantle was exhumed or an extreme thin layer of continental crust remained (Dannowski et al., 2020). The analysis of a seismic refraction study along a profile from the northern margin to the basin centre (Dessa et al., 2020) might shed further light on the crustal structure. The Corsica-Sardinia block underwent a counter-clockwise (CCW) rotation (Alvarez et al., 1973; Rehault et al., 1984;

Speranza et al., 2002; Maffione et al., 2008) of ~23° (Speranza et al., 2002) to 45° (Gattacceca et al., 2007) between ~21-16 Ma, and of 53° between 35-16 Ma (Le Breton et al., 2017), with an Euler rotational pole near Genoa, onshore or in the Gulf of Genoa (Fig. 2). Extension in the Ligurian Basin ended ~16 Ma and continued afterwards in the Algerian and Tyrrhenian basins (Mauffret et al., 2004).

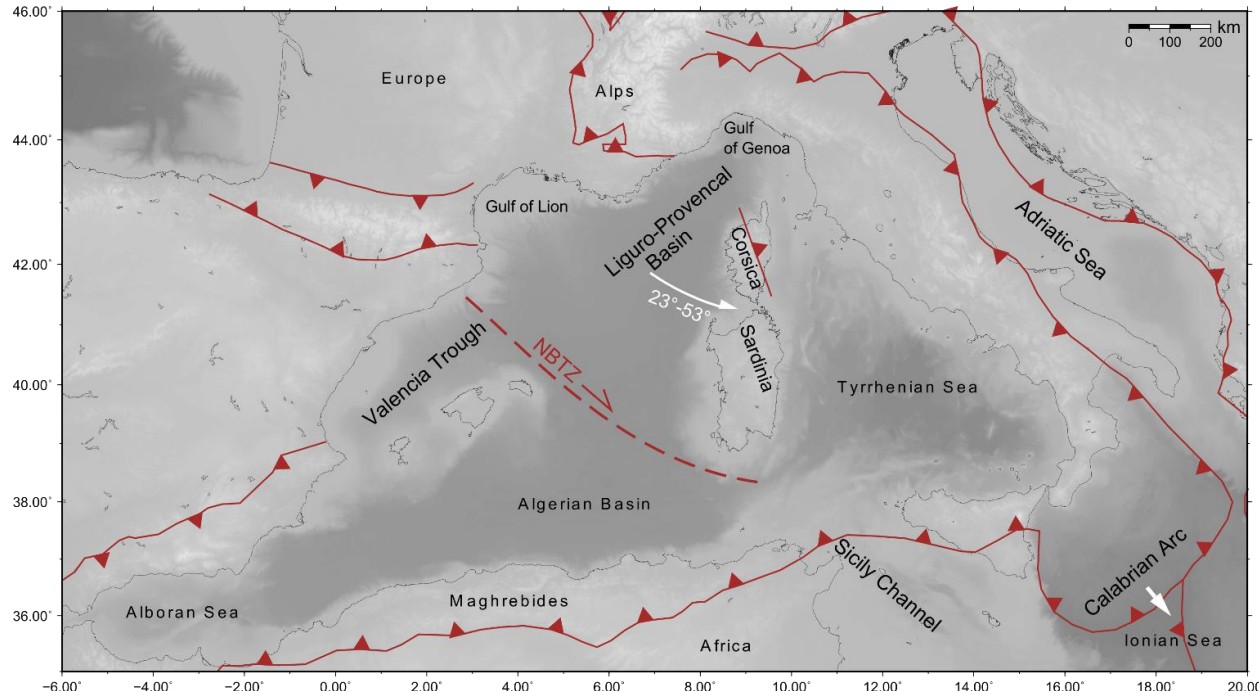

**Figure 2: Geographic and tectonic overview map with thrusts modified from Le Breton et al. (2020) and the North Balearic Transform Zone (NBTZ) (Hinsbergen et al., 2014). The white arrows indicate the ~23° to 45° CCW rotation of the Corsica-Sardinia block during Miocene times (Speranza et al., 2002; Gattacceca et al., 2007) and the ~53° during Oligo-Miocene times (Le Breton et al., 2017) and the trench retreat of the Calabrian Arc. GMRT data were used as background topography (Ryan et al., 2009).**

The Calabrian trench retreats further south-eastwards (Fig. 2, white arrow), but, recent seismological observations indicate that Tyrrhenian Sea opening slowed down or ceased and the Africa-Eurasia convergence results in basin inversion at its southern rim (Zitellini et al., 2020). It is proposed that the major shortening caused by the convergence between Africa and Europe is currently accommodated in the Maghrebides ranges in North Africa (90-100%) and that the Ligurian Basin and the Corsica-Sardinia block are rigid (Nocquet and Calais, 2004; Béthoux et al., 2008; Nocquet, 2012). However, passive seismic studies observe signatures of compression in the Ligurian Basin (Béthoux et al., 1992; Baroux et al., 2001; Eva et al., 2001; Courboulex et al., 2007; Béthoux et al., 2008; Larroque et al., 2011, 2012, 2016). The main portion of the compression is accommodated in active north dipping reverse faults at the Ligurian margin proposed to be active since ~5 Ma (Béthoux et al., 2008; Sage et al., 2011; Larroque et al., 2011). The Ligurian margin was uplifted by more than 1 km with respect to the basin (Sage et al., 2011; Larroque et al., 2011). Moreover, active compressional structures were imaged in seismic reflection profiles (Bigot-Cormier et al., 2004) offshore the Ligurian coast. Along the Corsica margin, seismic reflection data do not image significant faults that reached the surface in the last ~5 Ma, but earthquakes of $M_L$ 5.5 and $M_L$ 4.4 occurred in July 2011

offshore Corsica (Larroque et al., 2016) (Fig. 1, grey star). The low seismicity in the central basin and in the southern part
offshore Corsica indicate a weaker and more recent deformation (Béthoux et al., 2008; Larroque et al., 2016). While the
Ligurian margin is narrow and steep with a few listric normal faults, the Corsica margin is wider and several listric faults were
imaged in seismic reflection data (Finetti et al., 2005). Within a short distance of 30-50 km, the crust-mantle boundary (CMB)
deepens from ~15 km depth in the basin to ~25 km depth at the continental margins (Contrucci et al., 2001; Gailler et al., 2009;
Dessa et al., 2011).

## 3 Data and results

The AlpArray OBS network consisted of 24 broadband OBS (Fig. 1) that continuously recorded ground motion for ~8 months
(June 2017 to February 2018). The stations were deployed with a spacing of ~60 km using the French research vessel "Pourquoi
Pas?" and recovered during research cruise MSM71 on the German research vessel "Maria S. Merian". Data from two OBS
instruments types were used in our study: (1) German OBS, provided by the DEPAS pool (Schmidt-Aursch and Haberland,
2017) and GEOMAR (Lobster type), equipped with a Trillium Compact 120 s seismometer and an HTI-04-PCA/ULF
hydrophone recording on 6D6 KUM recorders with a sampling frequency of 250 Hz. (2) French OBS, provided by the IPGP-
INSU pool (BBOBS), equipped with a Trillium 240 s seismometer (T240) and a differential pressure gauge (DPG) recording
at a frequency of 62.5 Hz. Additionally, land data (Fig. 1) were used from different regional permanent and temporary
seismological networks: AlpArray (Z3), Italian National (IV), French RESIF-RLBP (FR) and Mediterranean MedNet (MN).

### 3.1 Events, picking and location

During the long-term OBS deployment, 39 seismic events were detected within the Ligurian Basin excluding the Alps-Liguria
junction zone (Fig. 1a, magenta circles). Our work focuses on two earthquake clusters in the centre of the Ligurian Basin, near
OBS A423A (magenta circles in Fig. 1b; Table 1). The first cluster (C1) consists of 13 events that occurred from June to
November 2017 (Fig. 3a, blue bars). The second cluster (C2) consists of three events that occurred during one day in January
2018, about 25 km southwest of C1 (Fig. 3a, red bars). Two events (grey bars in figure 3a) had few observations and high
uncertainties, these were not used in this interpretation. Two low magnitude events were only observed at station A423A using
the template matching method (e.g. Shearer, 1994) and were not further analysed (Fig. 3a, black bars). We show that a seafloor
network can detect more events than those in the land-based ISC catalogue (Tab. 1), whose magnitude of completeness is 2.2
in the region, but the four non-located events (grey and black bars in Fig. 3a) indicate that the AlpArray OBS station spacing
is too large to render a precise picture of the local seismicity.

We first used the ISC bulletin (Storchak et al., 2017) to identify seismic events in the AlpArray OBS data. The phases were
picked manually on all stations plotted with orange triangles, orange squares and white hexagons in Figure 1b. Only the
seismometer components were used in our analysis since the signal-noise ratio on most of the hydrophones was too low to
identify local seismic events.

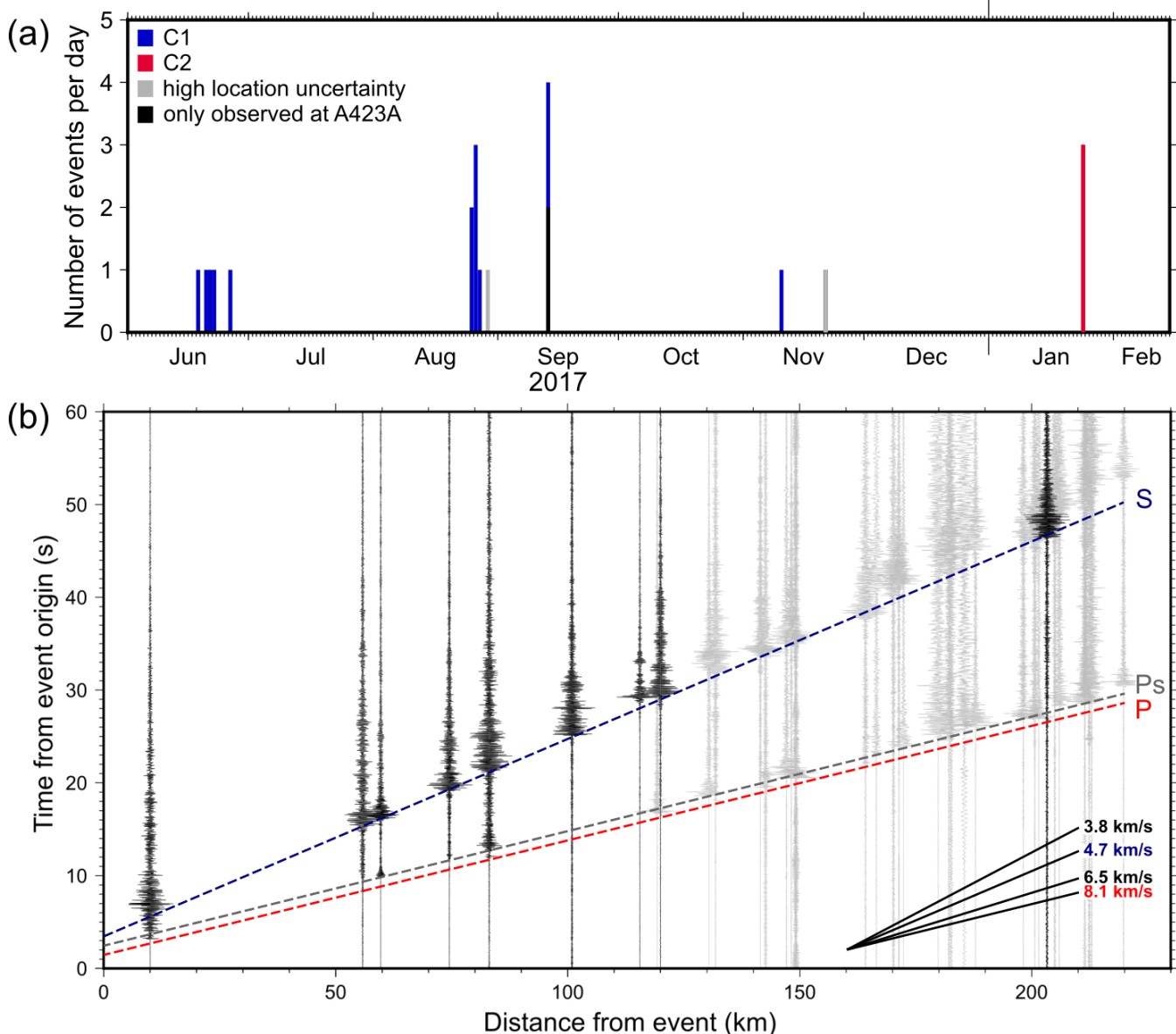

**Figure 3:** Panel (a) shows the temporal development of the cluster events in blue for C1 and in red for C2. Black events were only observed at A423A and not further analysed. Grey events were located with high uncertainties. (b) Waveforms of the strongest event of C1 (20 June 2017) displayed over offset. Black traces were used for EQ location. A weak P onset (red line) is observed, followed by a stronger Ps phase (grey line) ~1 s later. The S phase (blue line) is well visible at all stations. Apparent velocities of Vp=8.1 km/s and Vs=4.7 km/s are observed for the phase onsets (marked with dashed lines). More examples in Appendices (Fig. A1).

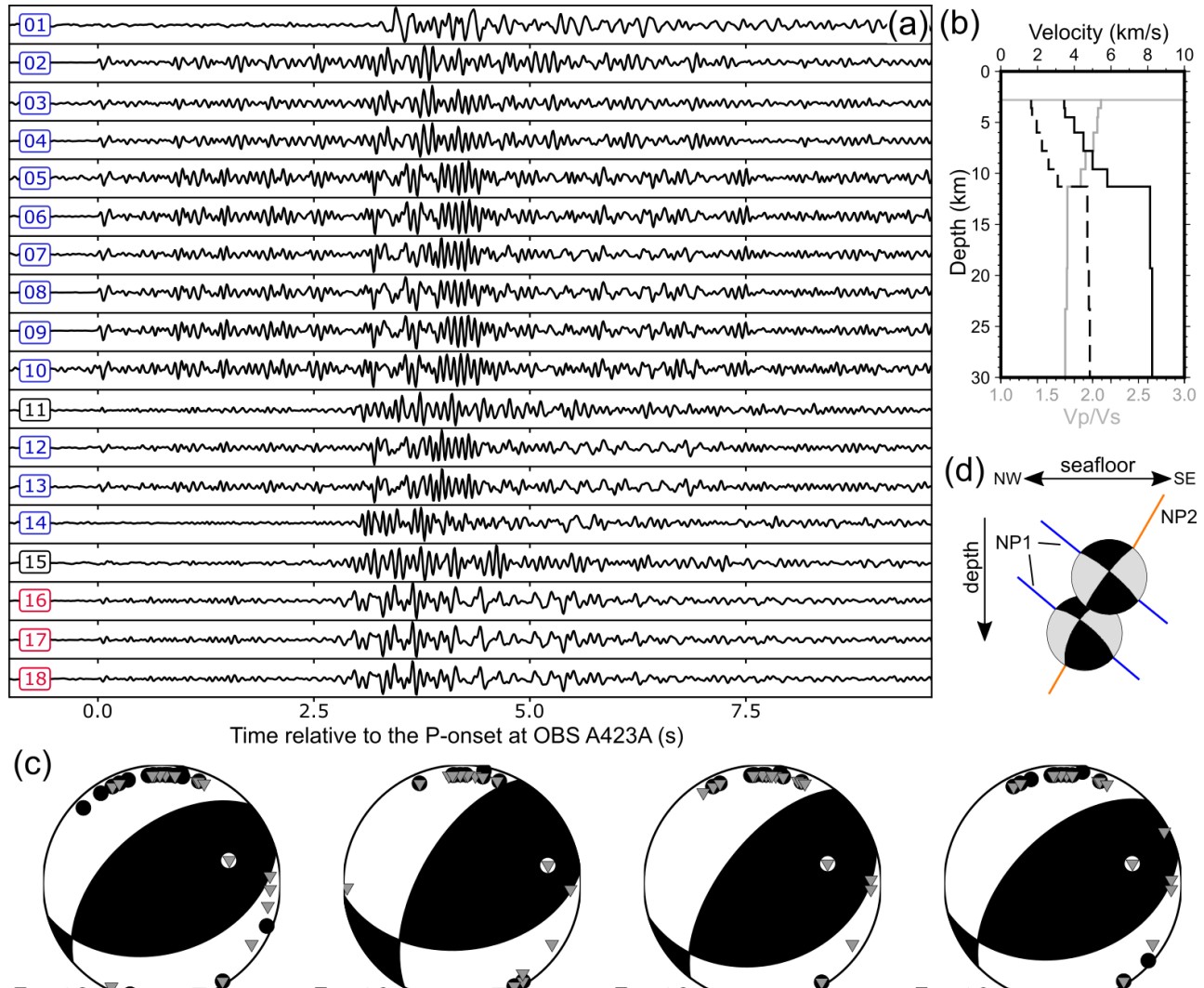

**Figure 4: (a)** All observed cluster events recorded at station A423A, Z-component (bandpass filtered 2-20 Hz). The P-onsets are shifted so that they align at 0 s. **(b)** 1-D velocity models (Vp, solid line, Vs, dashed line, and Vp/Vs grey solid line) used to relocate the cluster events. **(c)** Focal mechanisms of events 2, 6, 8 and 9, computed using the amplitude ratio of Sv/P (grey inverted triangles) and the wave polarisation (black dots for negative polarisation) determined at OBS and land stations. **(d)** Sketch of the two nodal planes (NP) of events 2 (upper) and 6 (lower) as side view.

Three onsets predominated the observed events: The P-wave, a converted Ps-wave and the S-wave (Fig. 3b, A1). The P-wave is weak in amplitude and followed by a stronger Ps-phase, which was observed on all OBS stations but not on land stations. S-wave amplitudes are increased by the seafloor itself due to the high impedance contrast. Additionally, the presence of a high or low velocity sedimentary layer with a high impedance contrast in the basin influences the wave field energy. This would only effect signals traveling through this layer, for example Messinian salts, towards the OBS and does not influence signals recorded on land stations. Because of these observations we use mainly amplitudes from land stations to estimate the fault

plane solutions. Both phases indicate an apparent P-wave velocity (Vp) of ~8.1 km/s and they are separated by a nearly constant
time difference of ~1 s. The S-wave phase has a high amplitude, compared to the P-onsets, and shows an apparent S-wave velocity (Vs) of ~4.7 km/s (Fig. 3b). The waveforms are highly coherent as shown for the vertical component of station A423A (Fig. 4a). We picked the P-onset on the vertical component and the S-onset on the horizontal components using SEISAN (Havskov and Ottemoller, 1999).

**Table 1: Epicentral locations and focal depths (with uncertainties of the relative depths) for the cluster events from HypoDD (depth below sea surface). Abbreviations: C – Cluster; WFF - waveform family; F – family.**

| Event ID | Date | Time of origin | Latitude | Longitude | Depth (km) | Magnitude ($M_L$) | ISC bulletin ID | C / WFF |
|---|---|---|---|---|---|---|---|---|
| 1 | 2017-06-18 | 20:55:29.48 | 42.194983 ± 0.0020 | 7.081169 ± 0.0020 | 10. 4 ± 0.2 | 1.2 | 610697551 | C1/- |
| 2 | 2017-06-20 | 17:09:51.36 | 42.203548 ± 0.0015 | 7.078639 ± 0.0015 | 15.5 ± 0.5 | 2.5 | 610697576 | C1/F1 |
| 3 | 2017-06-21 | 02:28:07.10 | 42.207711 ± 0.0037 | 7.077632 ± 0.0016 | 15.1 ± 0.7 | 1.2 | 610779982 | C1/F1 |
| 4 | 2017-06-22 | 08:06:29.84 | 42.201794 ± 0.0017 | 7.079651 ± 0.0015 | 14.9 ± 0.6 | 1.4 | 610780001 | C1/F1 |
| 5 | 2017-08-25 | 02:19:23.09 | 42.202787 ± 0.0022 | 7.069184 ± 0.0047 | 16.3 ± 0.4 | 1.3 | 611003163 | C1/F2 |
| 6 | 2017-08-25 | 02:33:43.42 | 42.202987 ± 0.0013 | 7.071468 ± 0.0015 | 16.1 ± 0.4 | 2.1 | 611003164 | C1/F2 |
| 7 | 2017-08-26 | 02:26:50.22 | 42.202734 ± 0.0018 | 7.065569 ± 0.0021 | 15.2 ± 0.6 | 0.9 | 611003237 | C1/F2 |
| 8 | 2017-08-26 | 02:30:33.03 | 42.201440 ± 0.0014 | 7.069776 ± 0.0015 | 16.1 ± 0.4 | 1.9 | 611003238 | C1/F2 |
| 9 | 2017-08-26 | 17:31:51.84 | 42.202738 ± 0.0014 | 7.070272 ± 0.0015 | 16.3 ± 0.4 | 1.9 | 610933407 | C1/F2 |
| 10 | 2017-08-27 | 00:39:30.31 | 42.200989 ± 0.0023 | 7.069771 ± 0.0042 | 16.0 ± 0.4 | 1.3 | 611003290 | C1/F2 |
| 11 | 2017-08-29 | 11:54:17.80 | 42.1850 ± 0.047 | 7.1040 ± 0.092 | 8.4 ± 2.9 | 1.1 | | - |
| 12 | 2017-09-13 | 07:47:16.31 | 42.205314 ± 0.0015 | 7.065561 ± 0.0015 | 16.5 ± 0.4 | 1.3 | | C1/F2 |
| 13 | 2017-09-13 | 09:26:57.97 | 42.207292 ± 0.0019 | 7.069896 ± 0.0017 | 15.9 ± 0.4 | 1.2 | | C1/F2 |
| 14 | 2017-11-10 | 23:05:48.52 | 42.204736 ± 0.0014 | 7.064332 ± 0.0016 | 14.4 ± 0.5 | 1.3 | 611617247 | C1/- |
| 15 | 2017-11-21 | 11:45:24.00 | 42.2080 ± 0.034 | 6.8180 ± 0.046 | 6.0 ± 5.2 | 0.9 | | - |
| 16 | 2018-01-24 | 05:11:25.24 | 42.089022 ± 0.0054 | 6.894502 ± 0.0050 | 9.9 ± 0.8 | 1.4 | | C2/F3 |
| 17 | 2018-01-24 | 07:35:48.27 | 42.089233 ± 0.0046 | 6.892409 ± 0.0074 | 9.9 ± 1.7 | 1.4 | | C2/F3 |
| 18 | 2018-01-24 | 09:43:10.11 | 42.091915 ± 0.0062 | 6.894325 ± 0.0051 | 10.5 ± 2.0 | 1.2 | | C2/F3 |

Within cluster C1, two main waveform families (WFF) are observed, indicating a repeated activation of the fault. Family 1 (F1) was active in June 2017 and consists of events 2, 3, and 4. Family 2 (F2) was active at the end of August and again in
September 2017 and consists of events 5 to 10, 12, and 13. Cluster C2 consists of a third waveform family (F3), events 16 to 18. The coherency within the families is >0.8.

We use only stations within the basin (orange triangles in Fig. 1) to locate the events. In this way, we can use a 1-D seismic velocity model for the basin and avoid errors introduced by the extreme topography and changes in crustal thickness near the margins. The determination of the absolute depths was challenging, since the stations show only Pn and Sn phases, except for
OBS A423A that was close enough to observe Pg/Sg and Pb/Sb phases. We applied a 1-D velocity model (Fig. 4b) based on P-wave velocities from seismic refraction profile p02 (Dannowski et al., 2020) (Fig. 5e). A Vp/Vs ratio of 2.0 at the seafloor

and 1.87 above the CMB was assumed to calculate Vs (Fig. 4b). The observed apparent velocities of mantle refracted waves Pn and Sn (Fig. 3b) were used as Vp and Vs for the uppermost mantle in the 1-D velocity models (Fig. 4b). First, an initial event location using HYPOCENTER (for event location) and RMSDEP (for uncertainties of absolute depths, Fig. A2) routines

within SEISAN (Havskov and Ottemoller, 1999 and references therein) was done. Afterwards, events of the two clusters were relocated with HypoDD, a double-difference earthquake algorithm for relative relocations (Waldhauser and Ellsworth, 2000). Because our clusters containing only a few events and the background velocity model is well-constrained, we used HypoDD's singular value decomposition (SVD) solver. All events of F1 and F2 (cluster C1) are located in the uppermost mantle (between 15 km and 17 km depth), while the events of F3 (cluster C2) are located in the crust. This is supported by observations at

station A423A, where Pg/Sg phases could be observed for the events of F3 but not for the events of F1 and F2. Further plots on the accuracy of the focal depths are given in the appendices (Fig. A2).

We compared epicentres of events observed in both the ISC bulletin and our study. Pairs are connected by black lines in the seismicity map (Fig. 1). The epicentres based on the ISC bulletin are spread over a 50x20 km zone, whereas the same events located solely using OBS data lie in a very narrow 3x3 km zone (Fig. 1b and 5b). Additional events with magnitude $M_L < 2$

were detected using the OBS; these events were also recorded by the land stations (Fig. 1) but their amplitudes were close to the noise level.

### 3.2 Focal mechanisms

To estimate fault plane solutions, we used the stations shown in orange in figure 1, plus the permanent station VSL in southern

Sardinia (located outside the map of Fig. 1a). The station distribution provided a good azimuthal coverage for the Ligurian Sea. was used. The first motion direction of the P-wave was determined for on- and offshore stations where clearly visible (examples in Fig. A3). The amplitude ratio of P- and S-wave was determined on the vertical component at land stations only. The amplitudes were corrected for attenuation effects using Qs=600 and Qp=1300 in the program FOCMEC (Snoke, 2003). The ocean bottom stations showed unusually small amplitudes for the P-wave compared to the S-wave, indicating a low

velocity contrast between sedimentary layers and water and suggesting a low subsurface shear modulus. Together with the high instrument mass, these effects could not be taken properly into account to determine amplitude ratios of P- and S-waves for the OBS. Polarity (Fig. A3) and amplitude ratios (Fig. A4) were used to derive the fault plane solution of events 2, 6, 8, and 9 (Fig. 4c; Tab. 2) using the program FOCMEC (Snoke, 2003). Strike directions and their uncertainties are presented in figure 4c. The uncertainties result from a systematic grid search for polarity and amplitude ratios using FOCMEC. Afterwards,

the 20 best possible solutions for each event were averaged and the standard deviation was calculated (Tab. 2). In general, the events of cluster C1 show stronger amplitudes compared to C2. The four fault plane solutions of cluster C1 indicate thrust faulting (Fig. 4c and Fig. 5a). The arrivals from events of C2 were not of sufficient quality to calculate focal mechanisms.

**Table 2: Fault plane solution for events 2, 6, 8, and 9.**

| Event ID | Date | Time of origin | Plane 1 | | | Plane 2 | | |
|---|---|---|---|---|---|---|---|---|
| | | | Strike | Dip | Rake | Strike | Dip | rake |
| 2 | 2017-06-20 | 17:09:51.36 | 74° +/- 5° | 41° +/- 2° | 110° +/- 5° | 229° +/- 5° | 52° +/- 2° | 74° +/- 4° |
| 6 | 2017-08-25 | 02:33:43.42 | 84° +/- 4° | 43° +/- 2° | 135° +/- 5 | 210° +/- 4° | 62° +/- 3° | 57° +/- 4° |
| 8 | 2017-08-26 | 02:30:33.03 | 73° +/- 29° | 33° +/- 10° | 121° +/- 9° | 217° +/- 25° | 63° +/- 9° | 71° +/- 8° |
| 9 | 2017-08-26 | 17:31:51.84 | 76° +/- 12° | 38°+/- 7° | 114° +/- 17° | 226° +/- 13° | 57° +/- 9° | 73° +/- 8° |

## 4 Discussion

### 4.1 Basin inversion

Geodetic measurements in Corsica and Sardinia show residual velocities <0.5 mm/y with respect to stable Europe (Nocquet and Calais, 2004), which the authors use to conclude that the shortening at the Alps-Liguria junction, expressed in widely

distributed low- to moderate-magnitude earthquakes, is a result of the ongoing CCW rotation of the Adriatic microplate, rather than a S-N motion of the Corsica-Sardinia block. Furthermore, Larroque et al. (2016) observed an earthquake cluster in 2011 offshore Corsica (Fig. 5a) accompanied by events in 2012 and 2013 located in the same area, but they did not observe surface ruptures nor faults rupturing the Plio-Quaternary sediments in the epicentre area of the 2011 events.

Inspired by their analysis, we studied the seafloor bathymetry in the C1/C2 region but neither observed any fault structures

nor do pre-existing multi-channel seismic data reveal sufficiently large faults in the sedimentary strata. There remains uncertainty on the depth of the base of the sediment layers from the refraction seismic study (Fig. 5e) (Dannowski et al., 2020), however, the C1 and C2 events are smaller than the 2011 events and occurred at similar focal depth. Thus, we assume that the rupture areas are entirely located within the lower crust and uppermost mantle and does not reach post-rift sediments. This suggests a long-term deformation without accumulating slip concentrated in one fault plane but rather distributed rupture areas.

C1 consists of two waveform families indicating repeated activation of the same fault plane for events of the same family. Events of one family have very similar waveforms (Fig. 4a) because they originate from the same fault plane. Events of family 2 occur at greater depth than events of family 1. We observe two possible nodal planes with a main strike in NE-SW to ENE-WSW (Fig. 4c and 4d, Tab. 2). However, we cannot identify which of the two was activated. For the second nodal plane (NP2, Fig. 4d) the event locations and the direction of the nodal plane coincide indicating that the same fault was activated at different

depths. For the first nodal plane (NP1, Fig. 4d) the events would have occurred on two neighbouring faults. The same is true for the relationship between C1 and C2, where we observed a third waveform family. Based on the data we cannot conclude if the clusters C1 and C2 belong to one fault or to two separate nearby faults, therefore we use the term 'rupture area'.

Larroque et al. (2016) debated whether the 2011 cluster results from ridge-push forces of an oceanic spreading centre in the Ligurian Basin or a southward propagation of the deformation at the Alps-Liguria junction. They exclude the hypothesis of a

spreading centre, since no spreading axis had been mapped. Our study corroborates this hypothesis, since clusters C1 and C2

are located in the basin centre and would be even closer to or at the proposed spreading axis and so ridge-push forces would be higher. Additionally, no spreading axis was mapped in previous seismic studies that interpreted the nature of the basin centre as 'atypical' oceanic crust (Contrucci et al., 2001; Rollet et al., 2002). Analysis of the LOBSTER seismic refraction profile p02 (Dannowski et al., 2020) proposes that rifting failed before seafloor spreading was initiated.

The driving mechanism for the deformation of the Ligurian basin has to be searched outside the basin. To summarise previous studies, sources for the regional compressional stresses in the basin centre could be: (1) Africa-Europe convergence, (2) CCW rotation of the Adriatic plate (Larroque et al., 2016), or (3) north-eastward motion of the Tyrrhenian Sea towards stable Europe (Nocquet, 2012). The geodetic network lacks stations in Northern Africa, excluding reliable geodetic constraints on plate motions (Nocquet, 2012).

The latest plate motion models (Nocquet, 2012) are based on seismicity and other geophysical and geological information and indicate that the majority (90-100%) of Europe-Africa convergence is accommodated in the Maghrebides. An analysis of two decades of dense GPS data presents a ~0.4 mm/y motion of Corsica representing a NNW-SSE shortening that is compatible with the tectonic and seismicity observations at the Ligurian margin (Masson et al., 2019). While Eva et al. (2020) show in a seismic study that the CCW rotation of the Adria block has no influence south of 45° N.

The epicentres of cluster C1 were located in the uppermost mantle (one event in the lower crust) and their mechanisms are thrust faulting (Fig. 5), while the epicentres of cluster C2 were located in the lower crust, above the CMB. The 2011 events were also thrust faulting events which occurred in the crust and uppermost mantle (Larroque et al., 2016), roughly 50 km E-SE of clusters C1 and C2. If we project C1 and C2 on line A-B that follows the push direction (based on the rake, Tab. 2) of the thrust events, they map in a slightly tilted vertical plane dipping north-westwards (Fig. 5d). Independently of the source of

regional stresses, we interpret the C1 and C2 clusters and the 2011 cluster as a result of basin inversion and hence a reactivation of the Oligocene-Miocene rift-related structures. The main portion of the basin inversion in the Ligurian Basin is accommodated at the northern margin where a high rate of seismicity is observed compared to the basin centre and the Corsican margin (Béthoux et al., 2008). Active northward dipping reverse faults have been mapped that are evidence for a 5 Ma cumulative deformation with a margin uplift of more than 1 km (Larroque et al., 2011, Sage et al., 2011). The centre of the

Ligurian Basin and the Corsican margin are characterised by low seismicity and diffuse distribution of rupture areas of small size spread over a wide area, which indicates the absence of cumulated deformation and points to a weaker or more recent deformation (Larroque et al., 2016). Shortening in the Ligurian Basin could reactivate these pre-existing rift-related structures, suggesting ongoing closure of the Ligurian Basin. Our results show that seismicity based on land stations underestimate the number of earthquakes in the Algerian and Ligurian basins. Reactivation of pre-existing and often rifting-related fault planes

was observed in other areas, for example in the Tyrrhenian Sea (Zitellini et al., 2020), in the Gulf of Cadiz (Grevemeyer et al., 2016) and in northern Honshu, Japan (Kato et al., 2009). Therefore, like the Tyrrhenian Sea (Zitellini et al., 2020), the Ligurian Sea may have entered a stage of basin inversion.

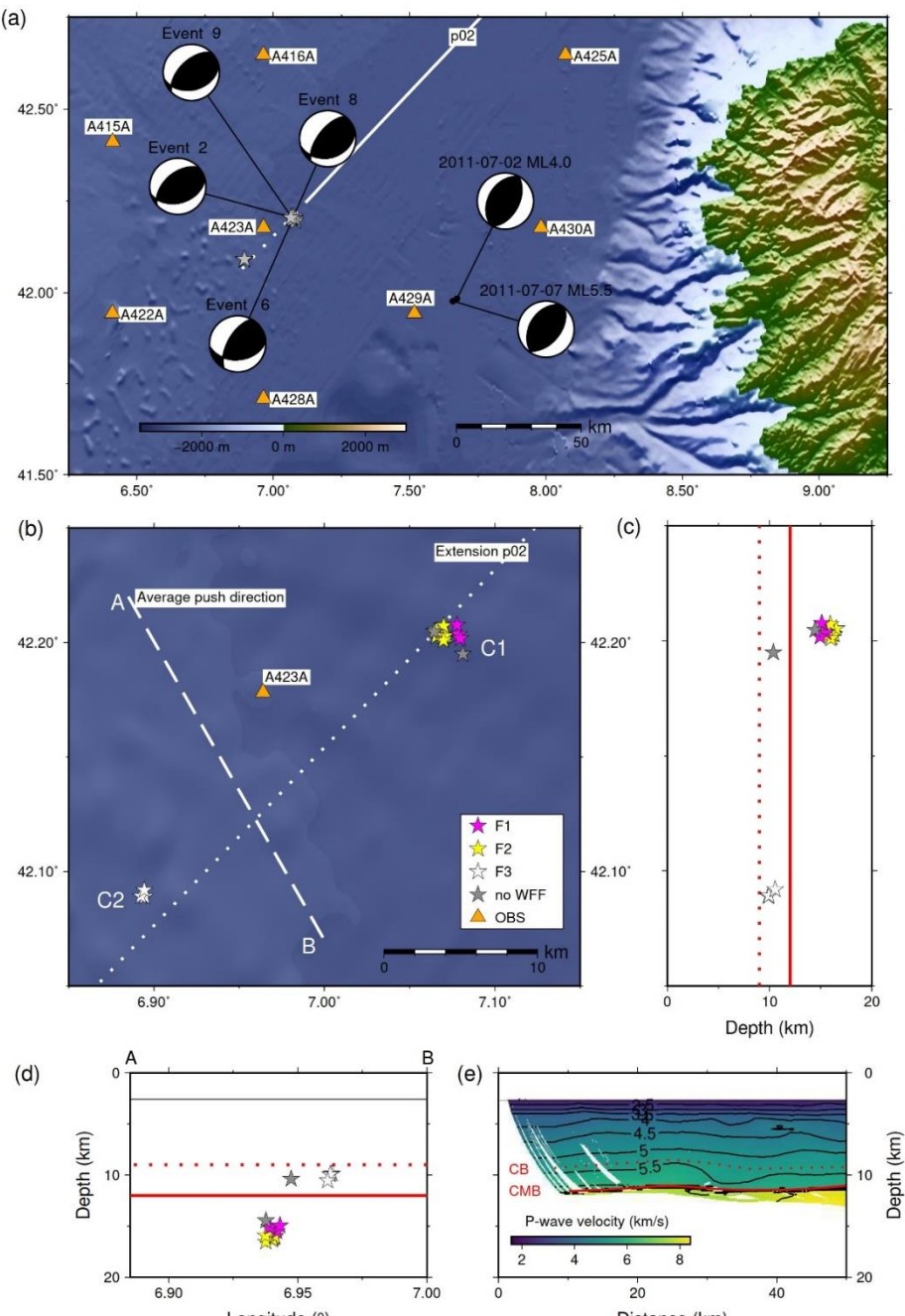

**Figure 5: (a) Focal mechanisms of C1 cluster and two events in 2011 (Larroque et al., 2016). (b) Zoom into the map from panel a. The white dotted line indicates the prolongation of seismic refraction profile p02 (white line in panel a) and the white dashed line represents the main horizontal direction of the pressure axis P as indicated by focal mechanisms. Events are displayed coloured according to their wave form family (WWF). (c) Depth distribution of the cluster events from North to South. (d) Events projected on the dashed line A-B shown in panel b. The thin black line indicates the seafloor, the dotted red line is the base of the sediment layers (CB) and the solid red line is the crust-mantle boundary (CMB). (e) SW end of the seismic velocity model computed from seismic refraction profile p02 (Dannowski et al., 2020). Topography from GMRT data (Ryan et al., 2009).**

## 4.2 Orientation of pre-existing rift-related faults

While earthquakes are spread over the entire Ligurian Basin (Fig. 1), the C1 and C2 events cluster within small areas. C1 and C2 are separated by ~25 km in a NE-SW direction. It is possible that both clusters may originate from the same fault zone, however, this cannot be clearly determined by our dataset. The focal mechanisms of cluster C1 are similar to the 2011 events (Larroque et al., 2016). We observe a difference in the average striking direction of ~40° for the first nodal planes and ~10° for the second nodal plane of C1 (WSE-ENE to SW-NE) compared to the striking direction of the 2011 events (SW-NE to SSW-NNE). As discussed before, we interpret the three clusters as caused by the reactivation of Oligocene-Miocene rift-related structures. Normal faults that were created during the extensional phase are inverted into reverse faults and their strikes are those of the normal faults during the extensional phase. Thermo-mechanical modelling suggests that rifting-related structures get younger oceanwards (Brune et al., 2014). Since the structures supported by the 2011 events are located more to the southeast, i.e. closer to the coast, they represent early rifting stage structures, whereas the structures supporting C1 and C2, located more to the centre of the basin, were developed during a later rifting stage. Thus, we speculate that this difference in strike (~10° or ~40°) might be connected to the CCW rotation of the Corsica-Sardinia block compared to stable Europe that was estimated in previous studies with ~23° (Speranza et al., 2002) to ~45° (Gattacceca et al., 2007) between 21-16 Ma or 53° between ~35-16 Ma (Le Breton et al., 2017) in total amount of rotation.

## 4.3 Rheology of crust and uppermost mantle

The occurrence of C1 at mantle depths and a high S-wave velocity combined with a low Vp/Vs ratio are puzzling. Earthquakes in continental domains normally occur in the upper crust, while the lower crust is relatively aseismic and earthquakes are rare in continental mantle lithosphere (Maggi et al., 2000). Whereas earthquakes certainly occur in oceanic crust and mantle lithosphere (Wiens and Stein, 1983). Similar to oceanic lithosphere, Maggi et al. (2000) suggest that the strength of continental lithosphere resides in one seismogenic layer within the crust and is controlled by the temperature structure and the amount of water in the crust. They propose that continental mantle lithosphere is relatively weak, and thus, aseismic. However, episodes of rifting may affect crustal and mantle rheology. On the other hand, Handy and Brun (2004) show that seismicity is not an indicator of rock strength but argue that seismicity may be used to locate active weak zones within continental lithosphere. Rifting models at non-volcanic rifted margins of the Atlantic-type suggest that the rheology of the continental domain changes during extension (Pérez-Gussinyé and Reston, 2001). With increased stretching, the portion of the crust that becomes brittle increases. Stretching of the lithosphere brings crustal rocks to lower pressure and temperature, and thus the lower part of the crust into brittle domain (Pérez-Gussinyé and Reston, 2001). When the entire crust is brittle, faults can cut through the crust into the mantle and act as fluid pathways, a pre-condition to initiate mantle serpentinisation during rifting (Pérez-Gussinyé and Reston, 2001). The serpentinites create a weak base of the crust, enabling detachment along the CMB and crustal fault block rotation. The serpentine thickness increases with increasing rift duration until the final break-up of the continent (Pérez-Gussinyé and Reston, 2001). The initial conditions and the evolution of the Atlantic-type rifting of old orogens differs from

the Ligurian Sea as back-arc basin where rifting took place during the alpine orogeny. Both margins show similarities and

differences: common features are highly weakened continental crust in the ocean-continent transition to a wide and thick basin starting rifting in subaerial conditions; the major difference is that in the Gulf of Lion the continent-ocean transition is probably made of exhumed lower continental crust, while in the Atlantic the upper crust rests directly on top of mantle (Jolivet et al., 2015). We recognise a good relationship between our data and the rifting model of Pérez-Gussinyé and Reston (2001), suggesting that the entire continental crust may have evolved into a more brittle domain during extension. Taking the depth of

the C1 events into account this suggests that the mantle was weakened, possibly due to local serpentinisation, down to at least 4 km depth below the CMB. While the high velocity Vs=4.7 km/s indicates a generally strong uppermost mantle.

The clustered events discussed here are interpreted to reflect inversion along pre-existing normal faults generated during rifting. Earthquakes within continental mantle lithosphere were also observed in the Gulf of Cadiz, in the southern Iberian old Jurassic mantle lithosphere (Grevemeyer et al., 2016). An explanation could be that during extension, mantle material moves

closer to the surface than before stretching, causing mantle temperatures to decrease (Sandiford, 1999). Thus, the mantle in the basin centre might become stronger and more brittle than the surrounding mantle (Sandiford, 1999). Such a scenario is supported by cluster C1 occurring in the uppermost mantle and supporting a low Vp/Vs-ratio of 1.72 (Fig. 4b). The crustal structure in the vicinity of clusters C1 and C2 is imaged by the LOBSTER seismic refraction profile p02 (Dannowski et al., 2020). Uncertainties remained for the depth of the base of the sediment layers and the thickness of the crystalline crust, while

the depth of the crust-mantle boundary is well imaged. The study provides no indication of a high amount of mantle serpentinisation at its southern end. A high sedimentation rate during rifting (Sage et al., 2011) may have prevented water from penetrating down to the mantle (Rüpke et al., 2013), minimising serpentinisation. However, water may have occasionally reached the mantle, causing serpentinisation around some rift-related faults, weakening the mantle locally (Pérez-Gussinyé and Reston, 2001). Thus, today it may enables the reactivation of rifting-related normal faults as reverse faults.

High heat flow (>100 mW/m²) in the Ligurian Basin (Pasquale et al., 1994) may contradict a cool CMB at the basin centre during rifting. However, recent studies indicate that the Ligurian Basin centre has not yet reached thermal equilibrium since the Oligocene-Miocene rifting process. Thermal modelling for sedimentary basins showed that a combination of a very thick sedimentary cover (>8 km), extremely low conductive sediments (<1.5 W/mK) and very shallow and localised crustal radiogenic heat production allow for a present day temperature maximum at the CMB beneath basin centres (Hansen and

Nielsen, 2002). Indeed, the thick sedimentary cover in the centre of the Ligurian Basin (up to 7 km, Schettino and Turco, 2006) might cause thermal blanketing, reducing the lithospheric heat loss, in line with the observed high heat flow values in the basin centre compared to the margins (Béthoux et al., 2008). These effects are also seen in thermal models from the Alps and their forelands, where temperatures at shallow depths (approx. 5 km below sea level) in the centre of the Molasse Basin are 20° C warmer than at the edges (Spooner et al., 2020). Further northeast of our study area, Béthoux et al. (2008) performed

2D thermomechanical modelling to understand the location of seismic activity. They show that the seismogenic zone in the centre of the Ligurian Basin reaches down to ~20 km depth. Béthoux et al. (2008) assume oceanic crust in the basin centre and

relate the location of earthquakes to contrasts in rheology and the presence of a continent-ocean transfer zone. The events of our observed C1 cluster range in the modelled seismogenic zone for the northern Ligurian basin; the contrast in rheology might be provided by pre-existing rift-related faults reaching the lithospheric mantle.

## 5 Conclusions

The entire Ligurian Basin is characterised by sparse but wide-spread micro-earthquakes of magnitude <3. The ocean-bottom seismometer deployment recorded between June 2017 and February 2018 two earthquake clusters that show thrust faulting mechanisms, supporting a model of inversion of the Ligurian Basin, in which the basin's centre is under compression and stresses are taken up by reactivated faults in the crust and uppermost mantle. Compressional forces are probably related to Africa-Europe plate convergence. The location of the cluster events and their focal mechanisms indicate that they occurred in reactivated pre-existing rift-related structures. Slightly different striking directions of faults in the basin centre compared to faults further east and hence away from the rift basin may reflect the counter-clockwise rotation of the Corsica-Sardinia block. In general, observations of earthquakes in continental mantle lithosphere are rare. A high mantle S-wave velocity of Vs=4.7 km/s and a low Vp/Vs ratio of 1.72 reveal a strengthening of the crust and uppermost mantle during the Oligocene-Miocene rifting. The observed event cluster indicate local weak zones possibly through local mantle serpentinisation in an otherwise strong lithosphere and support the interpretation that rifting failed in the northern Ligurian Basin. Additional data from an array of more densely spaced OBS would be needed to obtain a more complete picture of local seismicity in the basin centre.

## Appendices

A – Here we present additional data plots for data quality and location accuracy of the events that were used for the determination of fault plane solutions. Seismic sections (Fig. A1), theoretical and picked arrival times and RMS versus depth (Fig. A2), first motion plots (Fig. A3), and amplitude ratio (Fig. A4).

The software FOCMEC (Snoke, 2003) performs a systematic grid search over all possible the fault plane solutions. FOCMEC determines the polarity of the first motion and calculates the amplitude ratio between S- and P-wave for each fault plane solutions defined by the angles strike (0-360°), dip (0-90°) and rake (0-180°). Afterwards, these synthetic data (polarity and amplitude ratio) are compared with the observed data and reports the number of polarity errors and mean deviation of the amplitude errors. We selected the 20 best solutions for each event and determined the average fault plane solution. Figure A3 shows the quality of the first motion polarities and figure A4 shows the data fit between the observed and calculated amplitude ratios.

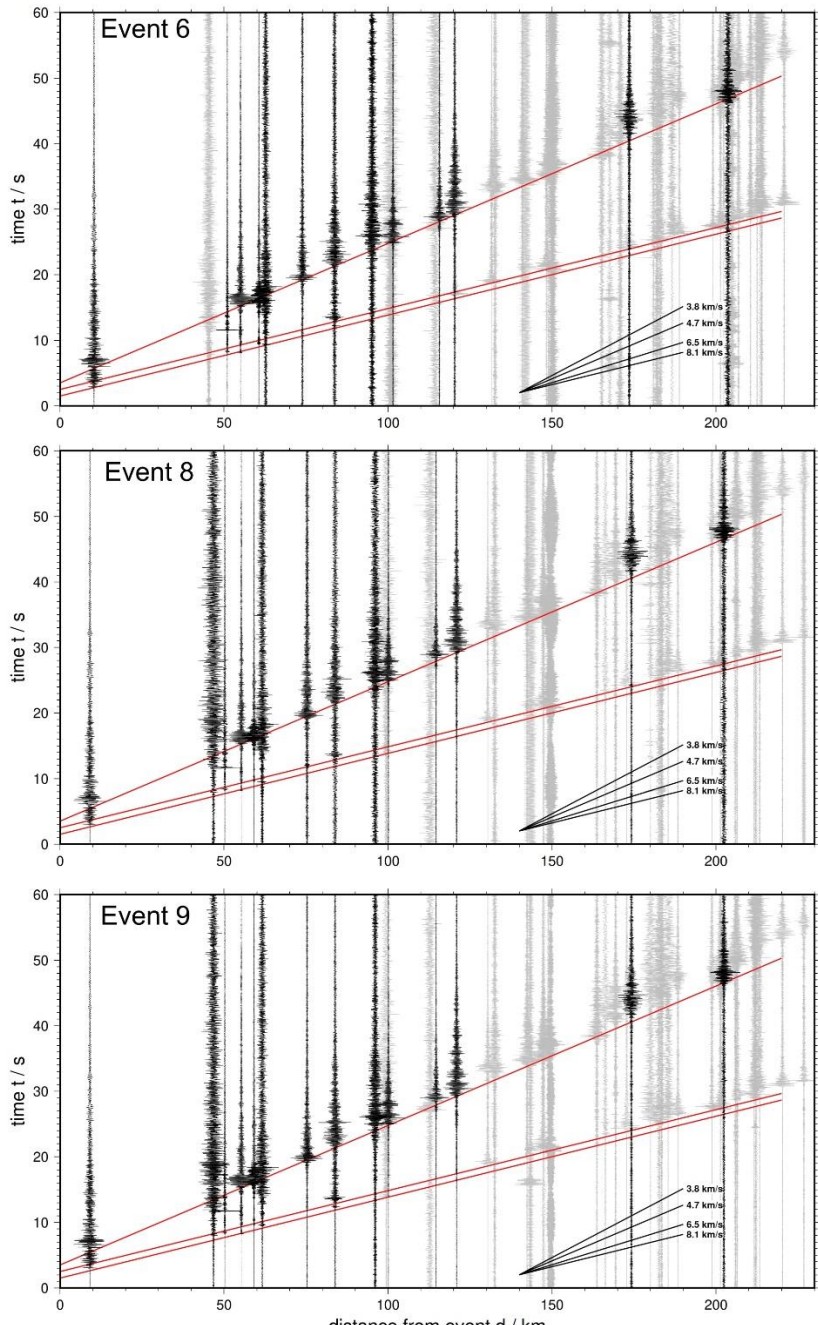


**Figure A1: Waveforms observed at stations displayed over offset. Black traces were used for earthquake location. Red lines mark the weak P onset, followed by stronger Ps and S phases. Apparent velocities of Vp=8.1 km/s and Vs=4.7 km/s observed for the phase onsets (marked with red lines).**

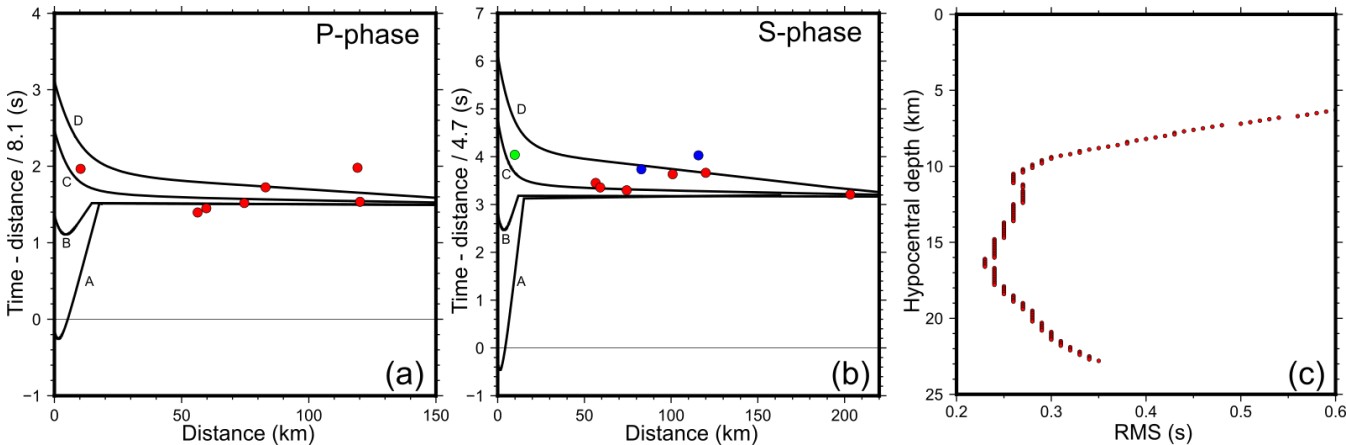


**Figure A2: Panels (a) and (b) show the theoretical (black lines) and observed (colored dots) first arrival times of the P- and S-phase for event 2. The color code displays the picking weight from SEISAN: red – 0 (best), green – 1, blue – 2. The theoretical arrival time were calculated for four different hypocentral depths: 6 km for upper crust (A), 10 km for lower crust (B), 16 km (C) and 20 km (D) both for upper mantle. Timing for A, B and D were manually adjusted so that the mantle phase is consistent in all cases. Panel (c)**
**displays the uncertainty (RMS versus depth) for the absolute depth estimation of event 2. SEISAN package "rmsdep" was used (Havskov and Ottemoller, 1999 and references therein).**

**Figure A3: First motion polarities for events 2, 6, 8, and 9 from left to right.**

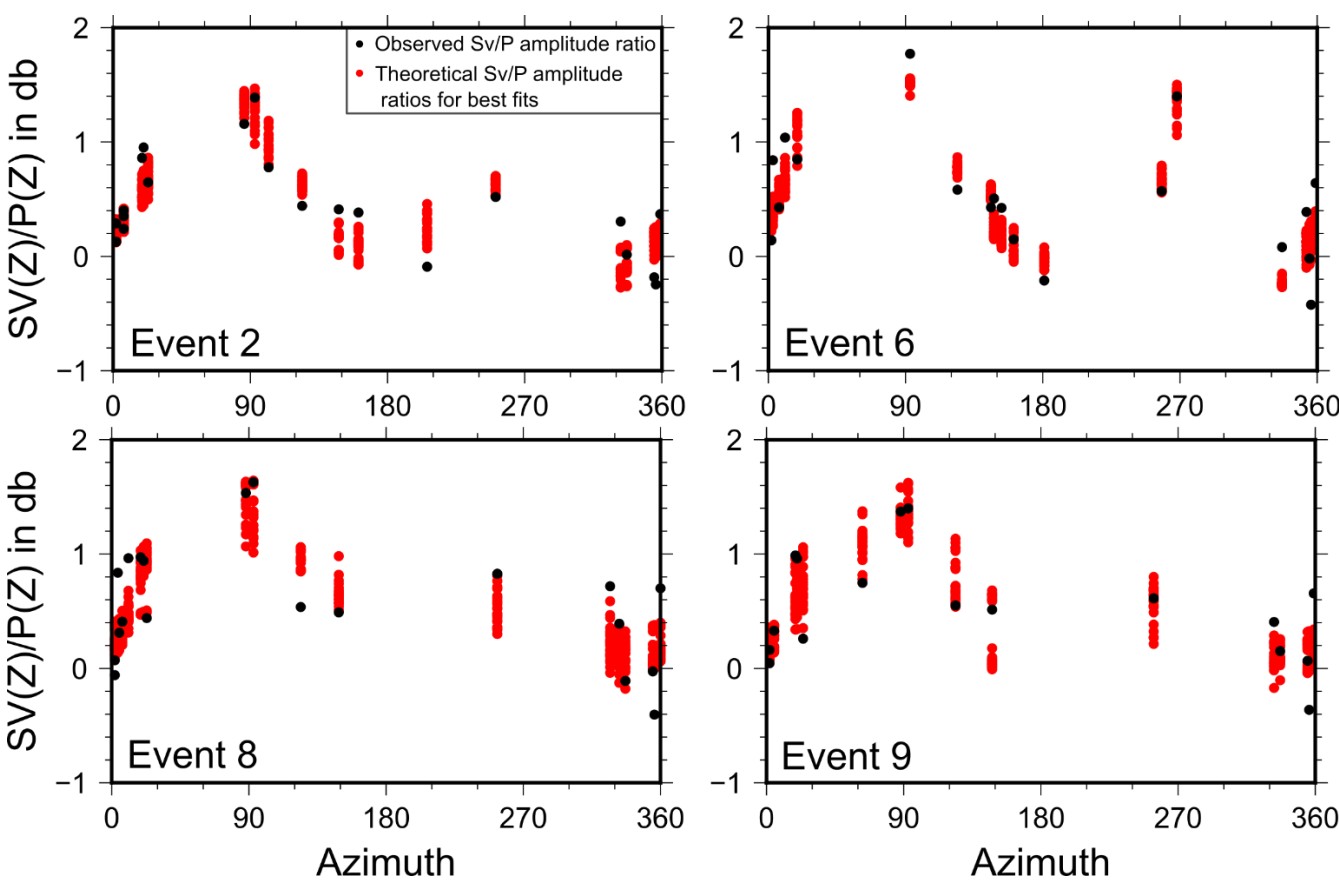

**Figure A4:** Amplitude ratios of Sv/P for the events 2, 6, 8, and 9. The calculated solutions for the fault plane solutions show a good fit with the observed Sv/P amplitude ratios.

**Acknowledgements**

We thank the captain and crew of the R/V Maria S. Merian (cruise MSM71, 7-27 February 2018) and the R/V Pourquoi Pas? (AlpArray cruise, 14-26 June 2017). We thank the cruise participants of both cruises for their efforts. OBS were provided by the DEPAS pool, GEOMAR, and IPGP-INSU pool. The deployment of the German component of the AlpArray seismic network (AASN) was funded by the LOBSTER project that is part of the German Priority Programme SPP2017 4D-MB. The deployment of the French component of the AASN was funded by the AlpArray-FR project of the Agence Nationale de la Recherche (contract ANR-15-CE31-0015). We also thank the AlpArray Seismic Network Team (http://www.alparray.ethz.ch/en/seismic_network/backbone/data-policy-and-citation/) and the permanent seismic networks used in this study (FR, IV, and MN). Figures were created using Generic Mapping Tools version 6 (Wessel et al., 2019), ObsPy (Beyreuther et al., 2010), and Inkscape (www.inkscape.org). We thank Christophe Larroque and Eline Le Breton for their reviews with very constructive comments and suggestions.

**Financial support**

This research has been supported by the Deutsche Forschungsgemeinschaft (DFG), grant no. TH_2440/1-1, KO_2961/6-1, and LA_2970/4-1. The deployment of the French component of the AlpArray seismic network (AASN) was funded by the AlpArray-FR project of the Agence Nationale de la Recherche (contract ANR-15-CE31- 0015).

**Data availability**

Data from the temporary and permanent land stations as well as the OBS are available through the AlpArray seismic network
(Z3, http://data.datacite.org/10.12686/alparray/z3_2015), the Italian network (IV, http://doi.org/10.13127/SD/X0FXnH7QfY), the French RESIF-RLBP network (FR, http://doi.org/10.15778/RESIF.FR), and the Mediterranean MedNet network (MN, https://doi.org/10.13127/SD/fBBBtDtd6q).

**Competing interests**

The authors declare that they have no conflict of interest.

**Author contributions**

MT carried out data analysis and contributed to the manuscript. AD wrote the manuscript and contributed to the data analysis. AD and MT created the figures. IG contributed to the manuscript with in depth discussion and manuscript editing. HK was PI of the research cruise MSM71 with R/V Maria S. Merian. WC was PI of the AlpArray cruise with R/V Pourquoi Pas?. HK, DL, IG, MT, AP, WC acquired funding and planned the research concept. HK, AD, FP, MT, DL, AP participated in cruise
MSM71 onboard R/V Maria S. Merian. WC, FP, AP deployed the AlpArray OBS network on board the R/V Pourquoi Pas?.

**Team list**

The complete member list of the AlpArray Working Group can be found at:

http://www.alparray.ethz.ch/en/seismic_network/backbone/data-policy-and-citation/.

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
