# Peer review of "Basin inversion: Reactivated rift structures in the central Ligurian Sea revealed by OBS"

_Solid Earth, 2021_

## Author Comment (AC1)

**Answers to reviewer RC1 Christophe Larroque**

We thank reviewer Christophe Larroque for the very helpful review and comments that in our opinion improve the manuscript.

We place the comments of RC1 in black and our answers and changes to the manuscript in green letters.

**Comments on the paper « Basin inversion: Reactivated rift structures in the Ligurian Sea revealed by OBS » by M. Thorwart et al., submitted to Solid Earth.**

**General comments**

The paper by Thowart et al. presents the seismotectonic interpretation of the data acquired in the Ligurian basin during 8 months in 2017-2018 by the 24 OBS of the Alparray experiment. This is the first data set of such importance acquired in this basin, with atypical oceanic crust or very thinned continental crust, located between the front of the southern subalpine chains and the Corso-Sardinia continental block. This area is known since a long time as an active intraplate setting with a very low deformation rate.

The experiment allows to detect 39 microearthquakes. This work focus on two clusters in the center of the basin, the first one with 13 events and the second with 3 events but only 4 focal mechanisms could be determined among the 13 events of the cluster 1. These 4 focal mechanisms, consistent with each other, are interpreted by the authors as evidence of the Ligurian basin inversion.

There is little new data but in this marine and low seismicity context any new data is welcome to be discussed and should be considered positively. The active inversion of the Ligurian basin has already been evidenced based on other structural and seismic datasets (Larroque et al., 2011; Sage et al., 2011; Larroque et al., 2016) and from this perspective this work confirms what is proposed. However, it seems to me that there are several important problems of substance and form in this article which require a serious reworking of the presentation of the data and their interpretation in the context of the work already carried out in this area. I recommend major revision.

**I make few remarks and suggestions in the following.**

- On the substance, the major problem concerns the exploitation of the seismic signal from very low magnitude earthquakes (1.5 to 2.5) recorded at stations located more than 150 km away (Fig1) to build the focal mechanisms. This essential part of the work must be strengthened to be credible. Firstly, as these are new mechanisms it is necessary to provide for each of them a clear diagram with the polarities and nodal planes (currently, only the diagram of the strongest earthquake is shown : c, fig4). Secondly, taking into account the large distances with some of the stations and the smallness of the magnitudes, the seismograms must be shown in order to attest the quality of the polarities read on this signal.

We updated figure 3c and provide a diagram for each event for which the focal mechanisms could be calculated. We included an Appendices as a new section in the manuscript and show there additional information on the 4 events with focal mechanism solution. This includes the seismograms, first motion plots and amplitude ratios of Sv/P.

- The context of the deformation in the Ligurian basin must be presented in its entirety. Particularly, the high rate of seismicity on the northern margin in relation to the center of the basin and its southern margin must be emphasized as it is an essential point in the discussion of the inversion process. From

this point of view, mention of the work of Béthoux et al (2008) is essential. Also from a structural point of view, it should be mentioned that active north dipping reverse faults have been identified on the northern margin (Larroque et al., 2011; Sage et al., 2011). These faults allow the accommodation of most of the basin inversion since 5 Ma, as evidenced by the cumulative deformation which shows a margin uplift of more than 1000 m with respect to the basin. These 2 points are important because they show that the basin inversion started at least 5 Ma ago in the northern part while the absence of cumulated deformation and low seismicity in the central and southern part (this paper and Larroque et al., 2016) attest to a weaker and/or much more recent deformation.

We included the works about the basin inversion of Larroque et al. (2011) and Sage et al. (2011) in the introduction section and explained better previous knowledge on the past and recent deformation of the Ligurian basin in section 2. And we modified section 4.1 according to the suggestions: "*The main portion of the basin inversion in the Ligurian Basin is accommodated at the northern margin where a high rate of seismicity is observed compared to the basin centre and the Corsican margin (Béthoux et al., 2008). Active northward dipping reverse faults have been mapped that are evidence for a 5 Ma cumulative deformation with a margin uplift of more than 1 km (Larroque et al., 2011, Sage et al., 2011). The centre of the Ligurian Basin and the Corsican margin are characterised by low seismicity and diffuse distribution of rupture areas of small size spread over a wide area, which indicates the absence of cumulated deformation and points to a weaker or more recent deformation (Larroque et al., 2016).*".

- The input of new data is really low for such a paper in a major scientific journal. May be the authors could try to get more informations from the continuous seismic recordings of the OBS by using template matching method ? Even if the signals are not usable to determine focal mechanics, it would be interesting to know if a larger number of low magnitude events can be detected.

We used the template matching method using a cluster event as template and observed 2 events near station A423A and displayed their occurrence in figure 3a with black bars (2 events in September). In this work we focus on the two cluster, thus we did not look for more low magnitude events near other stations. But this is indeed a good suggestion to go back to the entire dataset. We added the method in section 3.1 by changing the sentence: "*Two low magnitude events were only observed at station A423A **using the template matching method (e.g. Shearer, 1994)** and were not further analysed (Fig. 3a, black bars).*"

- You need to take better account of existing work when it relates to your interpretations. For instance, the proposal by Dannowski et al (2020) on the nature of the crust in the Ligurian Basin is very interesting but at the moment it is not a consensual result. So highlight other interprétations such as Contrucci et al. (2001), Rollet et al. (2002), Gailler et al. (2009).

We included the works of Contrucci et al. (2001) and Rollet et al. (2002) in the discussion section 4.1: "*Additionally, no spreading axis was mapped in **previous** seismic studies **that interpreted the nature of the basin centre as atypical oceanic crust (Contrucci et al., 2001; Rollet et al., 2002). Analysis of** the LOBSTER seismic refraction profile p02 (Dannowski et al., 2020) proposes that rifting failed before seafloor spreading was initiated.*". We did not include the work of Gailler et al. (2009) in this discussion since it is far away from the cluster.

- The use of the results of the Pérez-Guissinyé and Reston model does not seem to me very adapted to the case of the Ligurian rifting. This model has been proposed by its authors to describe a possible evolution of Atlantic-type rifted margins in the case of cratonic and old orogen models. It is therefore difficult to consider that this model is generalizable to all types of non-volcanic rifting. The Ligurian basin is a back-arc basin, the crust was strongly affected by the alpine orogeny. The initial rheological conditions are therefore strongly different from what is considered in the Pérez-Guissinyé and Reston model. This comparison should therefore be discussed and justified.

We stay with the comparison as a possible explanation for the observed events in the uppermost mantle. The differences between both margins in initial conditions and evolution do not exclude similar changes in rheology due to rifting. We emphasize that there are differences between both types of rifted margins by extending the discussion: "*The initial conditions and the evolution of the Atlantic-type rifting of old orogens differs from the Ligurian Sea as back-arc basin where rifting took place during the alpine orogeny. Both margins show similarities and differences: common features are highly attenuated continental crust in the ocean-continent transition to a wide and thick basin starting rifting in subaerial conditions; the major difference is that in the Gulf of Lion the continent-ocean transition is probably made of exhumed lower continental crust, while the in the Atlantic the upper crust rests directly on top of mantle (Jolivet et al., 2015).*".

**Specific comments**

- Revise the title because most of the inversion is active on the northern margin that is not discussed. For the moment, only a recent and low compression is carried out in the center of the basin.

We change the title to point out that the study is about the basin cenre: Basin inversion: "*Reactivated rift structures in the **central** Ligurian Sea revealed by OBS*".

line 27 : usually, moderate activity is less than magnitude 6 and strong activity start with Mw 6.5 thus the 1887 Ligurian eq (Mw > 6.5 Larroque et al., 2012, Manchuel et al., 2017) attests that the sismicity on the Ligurian margin is mainly moderate but with possible strong earthquakes (this is of concern for hazard assement).

We included the possibility for strong earthqueakes and added the reference: "*… indicating a moderate seismic activity **with occasionally strong earthquakes** (Béthoux, 1992; Courboulex et al., 2007; Béthoux et al., 2008; Larroque et al., 2012, 2016; **Manchuel et al., 2017**).*".

Line 51 : the geodynamic setting WAS controlled by the Africa/Euraisa convergence, now it is not so clear (see Nocquet and Calais, 2004, Serpelloni et al., 2007, ….).

Changed and references added

Line 54 : « Lamotte » is Frizon de Lamotte.

Changed

Line 70 : specify the rotation pole near Genoa and give the range of the counter-clockwise rotation from 23° (Speranza) to 45° (Gattacceca et al., 2007).

References added and changed to: "*The Corsica-Sardinia block underwent a counter-clockwise (CCW) rotation (Alvarez et al., 1973; Rehault et al., 1984; Speranza et al., 2002; Maffione et al., 2008) of ~23° (Speranza et al., 2002) to 45° (Gattacceca et al., 2007) or 53° (Le Breton et al., 2017) with an Euler rotational pole near Genoa, onshore or in the Gulf of Genoa (Fig. 2).*".

Line 159 : can you explain « faulty recording » ?

We changed "*faulty recording*" to "*a low velocity contrast between subsurface and water and might hint to a low shear modulus of the subsurface. Together with the high instrument weight, these effects could not be taken into account to determine amplitude ratios of P- and S-waves for the OBS*".

We did not take into account that even shear waves with a small amplitude would result in a large amplitude recorded at the OBS because the S-wave arrives at a free surface at the seafloor (doubling the amplitude), while the P-wave takes the seafloor as an interface with probably a low velocity contrast (little change in amplitude). Additionally the shear modulus of the subsurface might be small

and the mass of the OBS is large, resulting in a high S-wave amplitude. These effects should be taken into account for amplitude ratios.

Line 34 and 190-196 : I disagree with this presentation of the spatial geodetic data. Nocquet and Calais (2004) showed that the convergence of Africa relative to stable Europe is 40% less than the prediction of the Nuvel-1 model (De Metz et al., 1994). Serpelloni et al. (2007) and Nocquet (2012) argue that 90% of the convergence is accomodated along the Maghrebides chain and Algeria margin but ~10% of the motion could be accommodated northward with a possible NW motion of the Corsica–Sardinia bloc of 0.5 mm/yr maximum. Masson et al. (2019) confirmed a NW motion of Corsica in the range of 0.4 mm/yr.

*We added the new GPS work and adjusted the text as suggested in section 4.1: "An analysis of two decades of dense GPS data presents a ~0.4 mm/y motion of Corsica representing a NNW-SSE shortening that is compatible with the tectonic and seismicity observations at the Ligurian margin (Masson et al., 2019). It was proposed that this shortening is a result from the CCW rotation of the Adriatic microplate rather than from motion of an independent rigid Corsica-Sardinia block (Nocquet and Calais, 2004)."*

Line 194 : In my opinion reference to these long-term plate models (van Hinsbergen, Le Breton) are not relevant to discuss current movements because they have no resolution for the present-day.

*Removed*

Line 201 : the dip of the ~vertical plane (d, Fig5) is NW not SE ? In any case, in order to discuss the dip, one must have information on the uncertainties of epicenter location and depth because the difference are very tenuous.

*We added the horizontal uncertainties to the table. Changed to NW.*

Line 215-219 : you should mention the more recent and more precise work carried out on the rotation of the Corsica-Sardinia block by Gattacceca et al (2007). The rotation reconstructed from the paleomagnetic analyses of lavas is 45° counter-clockwise which leads to a much greater extension in the basin than from the 23° proposed by Speranza et al. (2002). At least both values should be mentioned.

*Changed to the total amount of rotation that varies depending on the different studies. We use now 23° to 53° and cite works of Speranza et al. (2002), Gattacceca et al. (2007) and Le Breton et al. (2017).*

Line 248-253 : the discussion about heat flow and the role of the sedimentary cover should take into account the results of Béthoux et al. (2008).

*Reference to Béthoux et al. (2008) added.*

Figure 2 : this figure is to be improved by showing the Ligurian thrust to the north, the normal faults of the Tyrrhenian Sea and the thrust to the north of Algeria….

*We added the NTBZ taken from Hinsbergen et al. (2014). We added the reference to the list. We removed "Balearic Sea". We do not add the suggested zones of inversion. We cited a number of studies that show and describe the zones of inversion in the western Mediterranean. The zones of inversion are wide and of different ages and we want to keep the map as simple and understandable as possible showing the main geological features and location of the basins we describe and discuss. Showing all the faults would need a zoom into the two basins.*

Caption figure 5 : (d) « crystalline basement » = top of crystalline crust ; number on (e) = Vp ?

Changed to *"top of crystalline basement"*; the numbers are P-wave velocities as given above in the colour bar.

**Technical corrections**

Check the references, e.g. Maggi et al. ???

Changed

Gattacceca, J., Deino, A., Rizzo, R, Jones, D.S., Henry, B., Beaudoin, B., Valeboin, F., 2007. Miocene rotation of Sardinia : new paleomagnetic and geochronological constraints and geodynamic implications. Earth Planet. Sci. Lett., 258, 359–377.

Included in the reference list

Larroque, C., Mercier de Lépinay, B., Migeon, S., 2011. Morphotectonic and fault–earthquake relationships along the northern Ligurian margin (western Mediterranean) based on high resolution multibeam bathymetry and multichannel seismicreflection profiles. Mar. Geophys. Res. 32 (1–2), 163–179, http://dx.doi.org/10.1007/s11001-010-9108-7.

Included in the reference list

Manchuel, K., Traversa, P., Baumont, D., Cara, M., Nayman, E. and Durouchoux, C., 2017. The French seismic CATalogue (FCAT-17). Bull. Earthquake Eng., 8, 16. 2227–2251, doi: 10.1007/s10518-017-0236-1.

Included in the reference list

Masson, C., Mazzotti, S., Vernant, P., 2019. Precision of continuous GPS velocities 20 from statistical analysis of synthetic time series. Solid Earth, 10, 329–342, https://doi.org/10.5194/se-10-329-2019.

We included instead: Masson, C., Mazzotti, S., Vernant, P., and Doerflinger, E.: Extracting small deformation beyond individual station precision from dense Global Navigation Satellite System (GNSS) networks in France and western Europe, Solid Earth, 10, 1905–1920, https://doi.org/10.5194/se-10-1905-2019, 2019.

Sage, F., Beslier, M.O., Thinon, I., Larroque, C., Dessa, J.X., Migeon, S., Angelier, J., Guennoc, P., Schreiber, D., Michaud, F., Stéphan, J.F., Sonnette, L., 2011. Structure and evolution of a passive margin in a compressive environment : example of the southwestern Alps-Ligurian basin junction during the Cenozoic. Mar. Pet. Geol., 28, 1263–1282, doi:10.1016/j.marpetgeo.2011.03.012.

Included in the reference list

Serpelloni, E., Vannucci, G., Pondrelli, S., Argnani, A., Casula, G., Anzidei, M., Baldi, P., Gasperini, P., 2007. Kinematics of the Western Africa-Eurasia plate boundary from focal mechanisms and GPS data. Geophys. J. Int., 169(3), 1180-1200, http://dx.doi.org/10.1111/j.1365246X.2007.03367.x.

Included in the reference list

---

## Author Comment (AC2)

**Answers to reviewer RC2 Eline Le Breton**

We thank reviewer Eline Le Breton for the review and her fruitful comments that in our opinion improve the manuscript.

We place the comments of RC2 in black and our answers and changes to the manuscript in green letters.

**Referee comment on "Basin inversion: Reactivated rift structures in the Ligurian Sea revealed by OBS" by Martin Thorwart et al., Solid Earth Discuss.,**

https://doi.org/10.5194/se-2021-9-RC2, 2021

This manuscript is a short paper presenting seismic activity in the Ligurian Sea, detected by the recently deployed AlpArray seismic network (offshore component LOBSTER), and therefore fits very well with this special edition of Solid Earth. The data and results are very interesting as it shows clusters of compressive earthquakes in the center of the Ligurian Basin, suggesting inversion of this part of the basin, and at intriguing high depths (10-16 km; lower crust - upper mantle). However, the authors need to better present/discuss the uncertainties of the method applied and their results, and improve the discussion of their results in terms of rift-related structures and strength of the lithosphere (see main comments below). Therefore, I suggest major revision.

**Main comments:**

**- Section 3 - Data, methods & results:** Section 3 lacks information on the uncertainties of the method applied and the results, especially regarding the depth of earthquakes and the orientation of the fault planes, which are essential to assess/support the interpretation in terms of rift-related structures and rheology that follows in section 4. Table 1 provides uncertainty range on the depth of each event but it is not explained how this was estimated?

*We added the horizontal uncertainties in Table 1 and inserted a second table with strike directions and uncertainties. Programs that were used to estimate the uncertainties are named in the text. We added the sentence in section 3.2: "Strike directions and their uncertainties are presented in figure 4c. The uncertainties result from a systematic grid search for polarity and amplitude ratios using FOCMEC. Afterwards, the 20 best possible solutions for each event were averaged and the standard deviation was calculated (Tab. 2)."*. *We added the section Appendices to provide more information on the data quality and results.*

**- Section 4.2 – Orientation pre-existing rift-related faults:** the title of section 4.2 should be changed. This studies provides insight on the present-day seismic activity of the basin but not on the rifting history. The authors discuss here the orientation of the faults obtained from the focal mechanisms of the earthquakes and that they interpret to be inverted, pre-existing rift-related faults. I agree, a change of strike from c. SW-NE to more SSW-NNE may reflect (not 'mimic') the rotation of normal faults formed during rifting during the CCW rotation of Corsica-Sardinia towards the SE relative to Europe. But this paragraph needs rephrasing and more information should be given, e.g. the exact strike (and dip?) of the fault planes obtained from the focal mechanisms. The authors should also quote papers concerning the evolution of continental rifting that shows younging of rifting-related structures towards the center of the basin (e.g. Brune et al. 2014, Nature communications) to support their interpretation. Also, the rotation of Corsica-Sardinia, was not only between 21-16 Ma but started with continental rifting at about 35-30 Ma, as indicated by the age of syn-rift sediments (e.g. Séranne 1999,

J. Geol. Soc. London; Jolivet et al. 2015, Tectonics; note the total amount of rotation between 35-0 Ma is estimated to a minimum of c. 53°, Le Breton et al. 2017). The phase between 21-16 Ma is often interpreted as the phase of oceanic spreading in the Liguro-Provencal Basin (e.g. Speranza et al. 2002; Le Breton et al. 2017), as suggested by the age of post-rift sediments along the margin of Gulf of Lion (e.g. Séranne 1999; Jolivet et al. 2015) and Sardinia (e.g. Sowerbutts and Underhill, 1998, J. Geol. Soc. London).

*We changed the title to "Orientation of pre-existing rift-related faults". We changed "mimic" to "reflect" in the abstract and conclusion. We added the reference to Brune et al. (2014) and added the sentence in the discussion to support our interpretation: "Thermo-mechanical modelling suggests that rifting-related structures get younger oceanwards (Brune et al., 2014)." We rephrased the section on the rotation angles and timing.*

**- Section 4.3 – Rheology of the lithosphere:** This section is important as the depth and location of the earthquakes in the upper mantle, in the middle of the basin is indeed quite intriguing. First, the author should better present the uncertainty on the nature of the crust and depth of crystalline basement (CB on Figure 5) in this part of the basin. Then, they discuss the seismicity in terms of strength of the lithosphere. However, Handy & Brun (2004, EPSL) discussed that seismicity is an ambiguous indicator of strength and proposed that earthquakes are more reasonably interpreted as a manifestation of transient mechanical instability within shear zones and may be used to locate active weak zones within the continental lithosphere. This should be mentioned and discussed here, as shear zones commonly form through lower crust-upper mantle during continental rifting (see for example Naliboff et al. 2017, Nature communications). Finally, when discussing the strength of the lithosphere, temperature is indeed an important parameter but the discussion here is not clear. Why does stretching of the lithosphere would cause "lower" temperature at the crust-mantle boundary (l. 229)? Stretching involves thinning of the lithosphere thus increasing of the geotherm, which is reflected by higher heat flow, as mentioned later in the text (l.248-253). This is contradictory and should be clarified.

*We added to the discussion the uncertainties of the model for the crystalline crustal thickness: "The crustal structure in the vicinity of clusters C1 and C2 is well imaged by the LOBSTER seismic refraction profile p02 (Dannowski et al., 2020)***. Uncertainties remained for the depth of the crystalline basement and the thickness of the crystalline crust, while the depth of the crust-mantle boundary is well imaged. The study*** provides no indication of a high amount of mantle serpentinisation at its southern end.".*

*We added Handy and Brun (2004) to the discussion which indeed helps to make our argumentation clearer. The seismicity is not reflecting strong lithosphere but rather weak zones in a possibly strong lithosphere. The high mantle S-wave velocity (Vs=4.7) and the low Vp/Vs ratio (1.72) argue for a strong lithosphere. We also rephrased the conclusions and abstract according to the new discussion: "A high mantle S-wave velocities and a low Vp/Vs ratio support the hypothesis of strengthening of crust and uppermost mantle during rifting-related extension and thinning of continental crust.".*

*We rephrased according to the reference to write clearer why crust can strengthen during extension: "Stretching of the lithosphere will cause ***cooling of rocks within the crust and result in a decrease of pressure***  at the crust-mantle boundary, which in turn, will strengthen the entire crust ***(Pérez-Gussinyé and Reston, 2001)***."*

**Specific comments:**

Abstract (lines 20-22) and Conclusions (lines 259-263) need to be improved/rephrased (see main comments above). For example, I would not say 'away from the abandoned rift' but away from the center of the rift basin; 'mimic' -> 'may reflect'

*Changed to "rift-related", "rift basin" and "may reflect"*

Line 35: the entire basin might be under compressive stress but the inversion is mostly observed along the margins of the Tyrrhenian Basin

*Rephrased: "… of the  basin **that is mainly observed along the margins** (Zitellini et al., 2020).".*

Line 45: 8 months – precise 2017-2018

*Added*

Line 52: rollback of the Apennines, Calabrian and Gibraltar subduction zones

*Added*

Lines 53-54: south-eastward migrating (or retreating) Apennines-Calabrian arc (Frizon de Lamotte et al.)

*Changed as proposed*

Line 56: Alboran Basin: c. 25 – 8 Ma (see for example Comas et al. 1999, Proceedings of the Oceanic Drilling Program, Scientific Result, Vol. 161)

*Changed and Reference added*

Lines 62-63: I would call it Liguro-Provencal Basin (SW part), not Balearic Sea. Gailler et al. 2009; Afilhado et al. 2015; Moulin et al. 2015 are geophysical studies between Gulf of Lion – Sardinia, which is the southwest part of the Liguro-Provencal Basin. And they mentioned "atypical" oceanic crust, it should be mentioned.

*Changed and added 'atypical'*

Line 70: Gattacceca et al. (2007, EPSL) should be also mentioned here.

*Added*

Lines 73-75: please add time constraints here, since when the opening of the Tyrrhenian Sea ceased?

*Added "**slowed down or** ceased" to express that it is very recent if it ceased.*

Lines 75-76: today

*Added*

Lines 180-181: The uncertainty on the depth of the crystalline basement (CB on Figure 5), especially for this part of the profile where the distinction between sediments and thinned continental crust cannot be done (Dannowski et al. 2020), should be mentioned and discussed (here and/or in section 4).

*Added here:"**There remains uncertainty on the depth of the crystalline basement from the refraction seismic study (Fig. 5e) (Dannowski et al., 2020), however, the**  C1 and C2 events …"*

Line 191: How does the counter-clockwise rotation of Adria would generate regional compression in the Ligurian Basin? Larroque et al. (2016) discuss the southward propagation of the deformation from

the Alps-Ligurian basin junction to the southern margin of the basin, for me this goes in (1) Africa-Europe convergence.

We only summarise the different observations of motion and this includes the rotation of Adria as a possible source for the compression of the two clusters.

Lines 193-194: Le Breton et al. (2017) and van Hinsbergen et al. (2020) are not really relevant here. Our plate reconstructions do not indicate where the Europe-Adria convergence is accommodated today, but provide information on the long-term plate motion and kinematics. They would be more relevant in geological setting, when presenting the geodynamic evolution of the area over the last 35 Ma.

We agree and removed both works in this context.

Line 195-196: what about the inversion of the northern margin of the Ligurian Basin? (as mentioned in section 2, see also Billi et al. 2011, Bulletin de la Société Géologique de France 182).

We added: *"An analysis of two decades of dense GPS data presents a ~0.4 mm/y motion of Corsica representing a NNW-SSE shortening that is compatible with the tectonic and seismicity observations at the Ligurian margin (Masson et al., 2019)."*

Sentences lines 182-183, 200-201 and 203-204: This is not very clear, do the authors suggest that these earthquakes occur along one fault plane (as projected in their profile AB, l. 200-201) or along different fault planes (as mentioned l. 182-183; 203-204)? And why (based on what arguments/observations)? The rupture lengths/areas must be small, as indicated by the low magnitude of these earthquakes, and may explain why the postrift sediments are not affected. But how does it tell more information on the location of theses earthquakes along one or more fault plane(s)?

We added an explanation to section 4.1 based on the waveform families observed in the data (section 3): *"C1 consists of two waveform families indicating repeated activation of the same fault plane for events of the same family. Events of one family have very similar waveforms (Fig. 4a) because they originate from the same fault plane. Events of family 1 occur at greater depth than events of family 2. We observe two possible fault planes (Fig. 4c, Tab. 2). For the second fault plane the event locations and the direction of the fault plane coincide indicating that the same fault was activated at different depths. For the first fault plane the events occurred on two neighbouring faults. The same is true for the relationship between C1 and C2, where we observed a third waveform family. Based on the data we cannot conclude if the clusters C1 and C2 belong to one fault plane or to two separate nearby fault planes, therefore we use the term 'rupture area' in the further discussion.".*

Line 200-201: 'push' direction -> slip direction (?)

Changed SE to NW

Line 204: 'can be taken up by these remaining' -> could reactivate pre-existing ; 'rifting structures' -> rift-related structures (check throughout the text); 'enabling' -> suggesting ongoing

Changed (throughout the manuscript)

l. 216: 'turned' -> inverted

Changed

l. 224: water in the formation of ?

Rearranged

l. 234: interpreted to reflect inversion along pre-existing normal faults

Changed

**Figures**

Figure 2: I would remove Balearic Sea and keep only Liguro-Provencal Basin (the authors could add the North Balearic Transform Zone, e.g. van Hinsbergen et al. 2014, Tectonics, as a delimitation between the Liguro-Provencal Basin and the Algerian Basin). I suggest also to add on this map the (inferred) location of oceanic crust (or atypical crust) vs exhumed mantle, and zones of basin inversion described in previous work and mentioned in the text.

We added the NTBZ taken from Hinsbergen et al. (2014). We added the reference to the list. We removed "Balearic Sea". We do not add the suggested zones of inversion and proposed location of oceanic crust. We cited a number of studies that show the outline of the oceanic crust and describe the zones of inversion in the western Mediterranean. The zones of inversion are wide and of different ages and we want to keep the map as simple and understandable as possible showing the main geological features and location of the basins we describe and discuss.

Figure 5: It would be interesting to plot the clusters of earthquakes directly on the seismic velocity model (e). At about 30 km distance, the velocity contour 5.5 km/s deepens, does the location of cluster C1 coincides with this change in velocity?

The deepening of the 5.5-isoline at the profile end is most probably an artefact resulting from lower resolution of the model at the end of the profile. We keep plotting the clusters beside the profile to prevent misunderstanding of the location of the cluster still enabling an easy comparison.

**References**

Bethoux 1992: Quaternaire (not quate)

Changed

Maggi et al. 2000: Journal (Geology) is missing

Added

---

## Referee Report (RR1)

Review of "Basin inversion: reactivated rift structures in the central Ligurian Sea revealed by OBS" by Thorwart et al.

First I would like to thank the authors for taking into account and replying to my comments during the first round of review. The uncertainties on the data and method section are now presented and discussed in more details. The manuscript is thus significantly improved. I have however some remaining points regarding the presentation/formulation especially of the amount of rotation of Corsica-Sardinia and in the discussion of the fault planes (4.1) and of the rheology/temperature/seismicity (4.3; see comments listed below) that would require some additional minor revision of the text.

**l. 19**: Oligo-Miocene rift basin
**l. 21**: Slightly different striking directions of presumed rift-related faults in the basin centre
**l. 23-24**: I find confusing to compare present-day S-wave velocities and Vp/Vs ratio to strengthening of the lithosphere during a rifting event that happened more than 16 Myrs ago. I would rephrase slightly here, following comments below on Section 4.3.

**l. 25**: which is no longer active and located in a plate interior.

**l. 74:** Careful here, there was a slight misunderstanding on the amount of rotation. Speranza et al. (2002) and Gattacceca et al. (2007), based on paleomagnetic data from Sardinia, suggest a rotation of ~23° to 45°, respectively, between ~21-16 Ma. In Le Breton et al. (2017), I estimated the total amount of divergence between southern France and Corsica-Sardinia since rifting started at about 35 Ma (so between 35-16 Ma, total amount of CCW rotation relative to Europe). This suggests a total rotation of at least 53° between 35-16 Ma (not only 21-16 Ma). So please remove "or" and rephrase for example as:
"a counter-clockwise (CCW) rotation relative to Europe (…) of ~23° (Speranza et al. 2002) to 45° (Gattacceca et al. 2007) between ~21-16 Ma, and of ~53° between 35-16 Ma (Le Breton et al. 2017), with …".

**Figure 2**: Either you remove the 53° from the Figure and keep the caption as it is, describing the CCW 23 degrees of rotation in Miocene times from Speranza et al. (2002). Or you should modify the caption accordingly to previous comment (53 degrees are for Oligo-Miocene times).

**l. 206-207, Table 1 and Figure 5**: Reading Table 1, C1/F1 are slightly shallower (~15 km) than C1/F2 (~16 km) not the other way around as mentioned in the text and colored on Figure 5. This should be double-checked.

**l. 207:** Please precise the main strike of the two planes (NE-SW to ENE-WSW) and that you cannot identify which of the two was the actual fault (activated/ruptured) plane as it didn't rupture the surface.

**l. 207-209**: "For the second fault plane …. For the first fault plane…" These sentences are not clear and should be rephrased. Do the authors mean rather the second / first family of events (and not fault plane)? An event location cannot coincide with a direction of plane…

**l. 211-212**: "therefore we use the term 'rupture area' in the further discussion", this part of the sentence could be removed as you don't really discuss the rupture area after that.

**l. 232-233**: "If we project C1 and C2 on line A-B that follows the push direction (based on the rake of Table 2 I suppose?) of the thrust events, they map in a slightly tilted vertical plane dipping north-westwards (Fig. 5d)."

I'm not sure to get what you want to say here: do you suggest then that this could indicate potentially the orientation of the ruptured fault plane? Would then the second nodal plane (Table 2), which has a higher dip (around 60°) and dip towards the NW (Table 2 and Figure 4) as along this A-B profile, be then the ruptured fault plane? Interestingly, this dip angle of 60° is typical for normal faults and you suggest in the following sentence that the earthquakes occurred along a pre-existing normal fault that was reactivated into a thrust. However, it could be quite steep for normal faults reaching down to lithospheric mantle.

**l. 251:** C1 (ENE-WSW to NE-SW)… 2011 events (NE-SW to NNE-SSW)

**l. 257-258:** I would either remove "that was estimated with ~23° to ~53° in total amount of rotation between 35-16 Ma" to avoid confusion as mentioned above in previous comment, or rephrase accordingly (23° is between ~21-16 Ma, 53° is for the entire rifting/spreading period ~35-16 Ma). You mentioned already the amount before so it might not be necessary to write it here again.

**Section 4.3:** In my opinion, it is important to clarify in this section processes that happened during rifting (35-16 Myrs ago) and the present-day state (thermal and rheology) that is important to understand present-day seismicity.
Most of the discussion - till line 301 - focuses on what happened during rifting. Indeed, it is important to mention it, to explain possible pre-existing weaknesses (rift-related normal faults) in the lithosphere and within the upper mantle, due to former rifting processes. These zone of weaknesses may be reactivated today due to regional compressional stress.

I would rephrase the sentence l. 272-273 to "Stretching of the lithosphere brings crustal rocks to lower pressure and temperature, and thus the lower part of the crust into brittle domain (Perez-Gussinyé and Reston, 2001)". Otherwise, it is confusing because during rifting, the thinning of the lithosphere rather increases, not cools, the overall temperature and high temperatures are usually associated with weakening of the lithosphere, not strengthening. But since rifting stopped, it must have cooled down since then (minus the effect of thermal blanketing from the thick sediment package that reduce lithospheric heat loss as mentioned).

This brings me to my last point, the last paragraph (l. 301-305) on heat flow and temperature is confusing/contradictory. Heat flow reflects the present-day thermal state, and this is crucial to understand depth of seismicity. Therefore, this part of the discussion should be improved.
The first sentence indicates that present-day heat flow is high in the basin centre, which may indeed contradict a cool CMB. The following sentence starts with "However" so we expect something explaining why the temperature may still be "cold" in the basin centre, but it finishes by "… allow for a temperature maximum at the CMB beneath the basin centre". So what is meant exactly here? Please clarify this paragraph.
Bethoux et al. (2008) is quoted at the end but is not discussed. But actually, this paper provide very good argument to explain the depth of seismicity. Indeed, their thermal modelling (their Figure 5) in the Ligurian Basin, although located more to the north and assuming oceanic crust in the basin centre, indicates depth of the isotherm 320°C, interpreted as seismogenic zone, ranging from 5 to 20 km across the basin. They mention the "occurrence of deeper (up to 20 km deep) earthquakes near the center of the basin, more likely favored by contrast in rheology", rheology contrast most likely due to the above mentioned pre-existing rift-related faults reaching the lithospheric mantle. Similarly, the recent work of Spooner et al. 2019 (Thermal field in the Alps and

its relation to seismicity; there's also Spooner et al. 2020 currently in revision in the same special issue of Solid Earth) discuss the link between thermal field and seismicity distribution in the Alpine area. They show a clear link between location of seismicity and depths of important isotherms within the continental crust/lithosphere (interpreted as mineral phase changes). Most seismic events occur between the 275°C and 450°C isotherms, which fits with the proposed seismogenic depth range in the modelling of Bethoux et al. (2008).

---

## Author Response (AR2)

We thank for the second review and for raising up further important points that easily could have misunderstood by the reader. We addressed all remarks and questions. We place the comments of RC1 in black and our answers and changes to the manuscript in green letters. We tracked changes in the manuscript.

Dear Editor,

The authors' response to the paper's review satisfactorily addresses many of the points I made (se-2021-9-AC1-supplement and se-2021-9-ATC1). Nevertheless, I think it is necessary to make some further clarifications and corrections in order to clearly precise some points and define the limits of the interpretations proposed in this work. I again recommend major revisions.
I address a new set of comments that are partly issue to discussion with my colleague Dr Marc Régnier (seismologist at Géoazur).
Sincerly,
C. Larroque

Main remarks on the manuscript se-2021-9-ATC1 by Thorwart et al. :
- Clarify the results of space geodesy regarding horizontal motions (these results, of course, have largely evolved since 20 years and the paper should highlight the updated interpretation) : some propositions are below.

We rephrased this section focusing on the centre of the basin and possible sources for compression.

- The analysis of the Ligurian basin is made from the structure proposed by Dannowski et al. 2020 considering that the central part of the basin is constituted by the hyper-thinned continental crust. It is quite legitimate that the authors rely on this hypothesis since it comes from their work and is argued by a published paper. This is not an issue to be re-discussed here but this point is important to be able to conclude, as the authors do, that the seismicity is localized on structures inherited from rifting. However, other interpretations of the central structure of the Basin at this location exist that propose the presence of exhumed mantle without hyper-thinned continental crust and this must be clearly mentioned (e.g. Canva et al., 2021).

Also the refraction seismic study published by Dannowski et al. (2020) leaves the question open, if mantle was exhumed or extremely thin continental crust remained. We added this extension to the sentence in section 2.

- The authors want to prove earthquake clusters in the mantle from location of epicenters and focal depths. The analysis of the seismological data currently presented is not sufficient to convince the reader : (i) the weak P-wave observed on figure 3B, for instance, is not discussed ; (ii) the absolute focal depths are not discussed taking into account the geometry of the network, the velocity model (…) ; (iii) in cluster 1 one event is observed above the moho and the others below: the comparison of the waveforms, above versus below, would allow to argue the focal depths in the mantle in the absence of absolute location.

(i) We added a short discussion on the weak P-wave signal in section 3.1: "*The P-wave is weak in amplitude and followed by a stronger Ps-phase**, which was observed on all OBS stations but not on land stations. S-wave amplitudes are increased by the seafloor itself due to the high impedance contrast. Additionally, the presence of a high or low velocity sedimentary layer with a high***"

*impedance contrast in the basin influences the wave field energy. This would only effect signals traveling through this layer, for example Messinian salts, towards the OBS and does not influence signals recorded on land stations. Because of these observations we use mainly amplitudes from land stations to estimate the fault plane solutions.*"

(ii) We added a figure in the appendices (new Fig. A2) showing the theoretical and observed first arrivals and a plot of RMS versus depth for event 2. We are confident that the hypocentres are in the mantle and that we observed also the mantle phase Pn/Sn on OBS A423A and not a crustal phase.

(iii) We added: "*All events of F1 and F2 (cluster C1) are located in the uppermost mantle (between 15 km and 17 km depth, while the events of F3 (cluster C2) are located in the crust. This is supported by observations at station A423A, where Pg/Sg phases could be observed for the events of F3 but not for the events of F1 and F2. Further plots on the accuracy of the focal depths are given in the appendices (Fig. A2).* " Further comments/discussion see next point.

In Fig. A2, the P- and S- phase arrive on OBS A423A later than a head wave travelling along the Moho discontinuity. This can only be explained by a hypocentre below the Moho (case C and D).

- The authors use HyppoDD to determine an uncertainty on the focal depth (some several hundred m, table 1) but this uncertainty is relative inside the cluster. However the authors discuss the absolute focal depths which is not correct because the uncertainty on the absolute focal depth is much larger and in such a context, with the closest station around 20 km from the cluster, may certainly reach sevreal km. The current text may create a major misunderstanding for the reader, this explanation must be reworked to point out that this is not an uncertainty on the absolute focal depths and some precautions must be taken in the further use of the values.

That is right, HypoDD determines uncertainties for relative depths. We added two sentences in section 3.1 that the depth estimation was difficult and that we used the SEISAN routine RMSDEP to check the absolute depths uncertainties: "*The determination of the absolute depths was challenging, since the stations show only Pn and Sn phases, except for OBS A423A that was close enough to observe Pg/Sg and Pb/Sb phases. … First, an initial event location using HYPOCENTER (for event location) and RMSDEP (for uncertainties of absolute depths, Fig. A2) routines within SEISAN (Havskov and Ottemoller, 1999 and references therein) was done. Afterwards, events …*".

- About the focal mechanisms and the use of FOCMEC : usually amplitude ratio for P-waves are determined on vertical component and amplitude ratio for SV and SH on horizontal component and SH are more likely to be reliable than those involving SV. As explained in the text, the authors determine the amplitude ratio of P- and S-wave on the vertical component only : do they consider that this have no influence on the result ? The use of the S/P ratio also requires the knowledge of attenuation factors (Qp and Qs) to correct the amplitudes, which is not mentionned. The use of the SV/SH ratio avoids this risky correction.

We added to section 3.2: "*The amplitudes were corrected for attenuation effects using Qs=600 and Qp=1300 in the program FOCMEC (Snoke, 2003).*"

Additionally, we determined the focal mechanism for event 2 using amplitude ratio between SV and SH both picked on the horizontal component. The resulting fault plane solution does not differ from our original solution. Therefore we stay with our original solutions.

- The focal mechanisms computed on figure 4 involve onland stations more than 100 km far from the source. What is the velocity model used to determine these focal mechanisms ? On figure 4 we observe rays from these onland stations with emergence angle of ~90° which is not realistic. The

authors must discuss these methodological issues show that their consequences do not call into question the results.

We used the velocity model displayed in fig 4b. It is correct that an emergence angle of ~90° for stations more than 100 km is not realistic for crustal events. But for events occurring in the upper most mantle it is. There, the ray paths of mantle phases Pn and Sn are more or less horizontal and only little curved, due to the small vertical velocity gradient in the mantle. Therefore, the emergence angle is close to 90°.

- The discussion about the rheology (4.3) need to be reworked, several inconsistencies make the text difficult to understand.

See comments below.

Some comments on the corrected paper (se-2021-9-ATC1) :

Line 21 (and also in the paragraph 4) : you mention "faults" but in reality it is the nodal planes of the focal mechanisms that are calculated. As you say in the article, potential faults are not identified and from the mechanism, not only can't one determine which of the two nodal planes is the fault plane (in this case, it will depend mostly on the dip of the fault) but also the direction is related to the quality of the mechanism and therefore to some uncertainty. I suggest to take care in the text of the paper.

We keep "*faults*" in the abstract but are careful in the discussion (nodal planes vs faults) as proposed from you later.

Lines 33-34 : « GPS data do not show any significant present-day shortening between Corsica and the northern rim of the Ligurian Sea (Nocquet and Calais, 2004) » and in the discussion you say that there is an horizontal motion measured by spatial geodesy ~0.4 mm/yr (Nocquet, 2012 ; Masson et al., 2019). It is certainly complex to explain the very weak motions observed by space geodesy and the evolution of these measures over the last 20 years in a few sentences. I suggest you simply say that Nocquet and Calais (2004) have shown that most of the Africa/Europe plates convergence is absorbed at the Maghrebian chain. Then, Nocquet (2012) and Masson et al. (2019) showed that a horizontal convergent motion exists between the Corsica-Sardinia block and the mainland Europe with a value of 0.4 mm/year.

Changed to: "*Nocquet and Calais (2004) have shown that the most of the plate convergence between Africa and Europe is accommodated at the Maghrebian chain. A present day horizontal convergent motion of 0.4 mm/year is observe between the Corsica-Sardinia block and mainland Europe (Nocquet, 2012; Masson et al., 2019). Compressive earthquakes …*"

Line 56 : « Serponelli » is better if Serpelloni.

Changed

Line 84 : Béthoux 1992 is Béthoux et al., 1992

Changed

Line 90 : « seismic rupture » is not the right term, seismic reflexion helps to image fault not seismic rupture.

Changed to "*faults*"

Line 175 : precise « subsurface sedimentary layers » ?

Changed

Line 192 : several times you mention « top of crystalline basement » may be precise what is it : continental crust-mantle-oceanic crust undifferentiated ? Wouldn't it be easier to talk about the sediment layer base ?

Changed to "*base of sediment layers*" at three places.

Lines 209-215 : difficult to understand may be a simple skecth would help the reader to follow your idea.

We added a sketch of the two different nodal planes as figure 4d and linked it in the text.

Lines 210-211 : « Events of family 1 occur at greater depth than events of family 2. ». If I look at the table 1, I read F1 : 15,5 km, 15,1, 14,9 and F2 : 16,3 km, 16,1, 15,2, 16,1…. Then F1 seems to be shallower than F2. Nevertheless I think that it is impossible to discuss such precision on the depth of microearthquakes located with stations more than 20 km away. It seems reasonable to me to delete this statement.

Exchanged 1 and 2. We keep it, depth uncertainties are given in the table. The uncertainties relative to each other should be good. Problematic might be the absolute depths. There was an error in figure 5 in the legend. It is now corrected to magenta star as F1 and yellow star as F2.

Line 211 : nodal planes are observed not faults. Following this observation you could interprete one of the nodal plane as the fault with some discussion.

Changed "*fault plane*" to "*nodal plane*" to discuss if one or two faults were active.

Lines 231-233 : « It was proposed that this shortening is a result from the CCW rotation of the Adriatic microplate rather than from the motion of an independent rigid Corsica-Sardinia block (Nocquet and Calais, 2004). » If you discuss the question of the origin of the compression on the northern Ligurian margin, then it must be presented in its entirety because it is obviously an important point for the question of inversion even if it is not completely resolved at present :
- Nocquet and Calais (2004): at that time measurements were not sufficient to detect the weak horizontal motion between the Corsica-Sardinia block and the mainland. They suggested that the counterclockwise rotation of Adria could be the cause of this compression, although the rotation pole near Turin strongly limits this influence.
- Larroque et al (2009) propose that this compressional zone at the southwestern junction of the Alpine-Ligurian Basin could be related to the stress generated by crustal thickening of the chain to the north with also as a consequence the extensional tectonic regime on the high peaks of the southern Alps.
- Sanchez et al (2010) question this extensional regime in the Alpine domain and propose that the Adria rotation controls the deformation up to the northern margin of Liguria.
- Finally Eva et al (2020) show from seismicity analysis that the counterclockwise rotation of the Adria block has no influence south of 45°N and thus no influence on the compressive regime of the northern Ligurian margin (which is confirmed by the analysis of extensive earthquakes over the Southern Alps [(Thouvenot et al, 2016) and GPS measurements over the Southern Alps (Mathey et al, 2020)].

The current consensus solution is the horizontal northward displacement toward NW of the Corsica-Sardinia block.

We took away the sentence and added Eva et al. (2020) to the discussion. We wanted discuss sources for compression of the basin centre, not at the Ligurian margin, since C1 and C2 are located in the basin centre: "*To summarise previous studies, sources for the regional compressional stresses **in the basin centre** could be: (1) Africa-Europe convergence, (2) CCW rotation of the Adriatic plate (Larroque et al., 2016), or (3) north-eastward motion of the Tyrrhenian Sea towards stable Europe (Nocquet, 2012). The geodetic network lacks stations in Northern Africa, excluding reliable geodetic constraints on plate motions (Nocquet, 2012). The latest plate motion models (Nocquet, 2012) are based on seismicity and other geophysical and geological information and indicate that the majority (90-100%) of Europe-Africa convergence is accommodated in the Maghrebides. An analysis of two decades of dense GPS data presents a ~0.4 mm/y motion of Corsica representing a NNW-SSE shortening that is compatible with the tectonic and seismicity observations at the Ligurian margin (Masson et al., 2019). It was proposed that this shortening is a result from the CCW rotation of the Adriatic microplate rather than from the motion of an independent rigid Corsica-Sardinia block (Nocquet and Calais, 2004). **While Eva et al. (2020) show that the CCW rotation of the Adria block has no influence south of 45° N.**"*

Line 254 : concentration of earthquakes was already mentionned in the ligurian domain : on the northern margin and in the 2011 epicentral area.

We keep this section as it is to have a short introduction into the new chapter.

Line 255 : the hypothesis that C1 and C2 occurred on the same fault because they area as close as 25 km is … an hypothesis : take precautions and be less assertive.

Changed to: "*It is possible that both clusters may originate from the same fault zone, however, this cannot be clearly determined by our dataset.*"

Line 255-265 : the discussion about the difference in orientation is interesting but donc forget that you discuss about nodal planes and the faults are not identified (same topic as line 211). Could the authors propose some hypothesis on which one of the nodal plane is the fault plane ?

Changed first and second "*fault plane*" to "*nodal plane*" according to line 211.

Line 262-263 : « Since the 2011 events are located more to the southeast, closer to the coast, they represent an older phase of rifting compared to C1 and C2 » may be replace by : Since the structures supported the 2011 events are located more to the southeast, i.e. closer to the coast, they represent early rifting stage structures whereas the structures supporting C1 and C2, located more to the center of the basin, were developped during a later rifting stage.

Changed

Line 267 : take precaution about the location in the mantle as the absolute uncertainties on the focal depth is of several km.

For the location of the EQ we used only stations in the Ligurian Basin to avoid 3D effects from topography or thicker crust. Absolute depth still can have high uncertainties, however, we observe a high apparent P- and S-wave velocity that confirms our interpretation that the earthquakes occur at mantle depth.

Line 272 : « formation » : could you precise what is it, I think it is the amount of water in the crust ?

Changed to "crust".

Line 273-276 : I don't understand the discussion about Handy and Brun (2004) : the seimicity is not an indcator of rock strength but it concentrates in weak zones but the weak zones are indeed zones of low resistance no ? This is confusing for me.

In strong zones the Earth would not break. Weak in the sense possible to break but not so low resistant that tensions are discharged constantly.

Line 277-280 : I agree the remark of Dr Le Breton about the section 4.3 and the new formulation of the authors leaves me confused. Currently, heat flow measurements through active rift such as the East African display high heat flow then a warming of the crust compared to the standart situation. Heat flow in the Ligurain basin (Della Vedova et al., see discussion in Bethoux et al., 2008) is currently high. I don't understand how the crustal thinning could allow a cooling of the temperature in crust ? nor the role of the drop of pressure on the CMB in the strengthenning of the crust ?

We rephrased this section to clearly differentiate, that heat flow values reflect the present day thermal state of the basin not the state during the rifting process. And we added more discussion on results of Béthoux et al. (2008) as proposed by Le Breton.

Line 287 : what is « attenuated crust » ?

Changed to "*weakened crust*"

Line 289 : IN THE Atlantic rather than « …the in the… »

Changed

Line 293 : « 4 km depth » precise : depth below sea level, below the CMB ?

Added "*below the CMB*".

Line 292-294 : you propose that extension developped without stretching ? It seems to me that the 2 two processus act jointly ? How to produce normal faults without a minimum of stretching ?

Added a "more" to the sentence. Both processes take place coincidently.

Lines 309-314 : Definitely there is something to rethink in the presentation of these interpretation about the rheology. This paragraph is clearly contradictory with the previous. You propose here that the crust is hot whereas you propose previously that the crust is cold (line 277 : « cooling of rocks within the crust »). Furthermore the présentation of a high heat flow in the central Ligurian basin is rather too simple and misleading. First you must mentionned that the work of Hansen and Nielsen is not dedicated to the Ligurian basin but is a modelling of sedimentary basins in general in order to study the relationships between lithospheric structures and (permanent relative weakness zones) and thermal structure and they propose : « the maximum Moho temperature, and therefore also the weakest upper mantle, is encountered beneath the flanks of the basin » and not « …localised crustal radiogenic heat production allow for a temperature maximum at the CMB beneath the basin centre. » as proposed by the authors. In the following they explain why the sediments thermal blanketing could be responsible for the weakest crust in the center of the basin. The modeling proposed by béthoux et al (2008) also deserves to be presented in a little more detail if you want to argue your interpretations : what about temperature pattern, the heat flow (observed and rougly corrected by Della Vedova, Pasquale…) and calculated in the Ligurian basin ?

Added "Thermal modelling for sedimentary basins" to Hansen and Nielsen (2002). We rephrased this section, included the timing (rifting time and present day) and extended the discussion taking comments from Le Breton into account, especially on the results and draw conclusion to our results.

Line 341-343 : in the legende of A2 precise « first motion polarities for events 2, 6, 8 and 9 from left to right ».
Added

Canva et al. Structural inversion of the North Ligurian margin: results from the SEFASILS experiment, EGU General Assembly 2021, online, 19–30 Apr 2021, EGU21-9759, https://doi.org/10.5194/egusphere-egu21-9759, 2021.

Larroque, C., Delouis, B., Godel, B., Nocquet, J.M., 2009. Active deformation at the southwestern Alps–Ligurian basin junction (France–Italy boundary) : evidence for recent change from compression to extension in the Argentera massif. Tectonophysics, 467 (1–4), 22–34, http://dx.doi.org/10.1016/j.tecto.2008.12.013.

Both works not added, we keep the discussion on the central basin and do not extent it to the Ligurian margin.

Eva, E., Malusà, M.G., Solarino, S., 2020. Seismotectonics at the Transition Between Opposite-Dipping Slabs (Western Alpine Region). Tectonics, 39, https://doi.org/10.1029/2020TC006086.

Added

Thouvenot, F., Jenatton, L., Scafidi, D., Turino, C., Potin, B., Ferretti, G., 2016. Encore Ubaye : Earthquake Swarms, Foreshocks, and Aftershocks in the Southern French Alps. Bull. Seism. Soc. Am., 106, 2244–2257, https://doi.org/10.1785/0120150249.

Not added, we keep the discussion on the central basin and do not extent it to the Ligurian margin.

Mathey, M., Walpersdorf, A., Sue, C., Baize, S., Deprez, A., 2020. Seismogenic potential of the High Durance Fault constrained by 20 yr of GNSS measurements in the Western European Alps. Geophys. J. Int., 222, 2136-2146, https://doi.org/10.1093/gji/ggaa292.

Not added, we keep the discussion on the central basin and do not extent it to the Ligurian margin and the Alps.

We thank for the second review and for raising up further important points. We think that especially section 4.3 is now improved and clearer to the reader. We addressed all remarks and questions and are very grateful for the recommended changes to the manuscript. We place the comments of RC2 in black and our answers and changes to the manuscript in green letters. We tracked changes for both reviewer comments (RC1 and RC2) in the manuscript.

**Review of "Basin inversion: reactivated rift structures in the central Ligurian Sea revealed by OBS" by Thorwart et al.**

First I would like to thank the authors for taking into account and replying to my comments during the first round of review. The uncertainties on the data and method section are now presented and discussed in more details. The manuscript is thus significantly improved. I have however some remaining points regarding the presentation/formulation especially of the amount of rotation of Corsica-Sardinia and in the discussion of the fault planes (4.1) and of the rheology/temperature/seismicity (4.3; see comments listed below) that would require some additional minor revision of the text.

l. 19: Oligo-Miocene rift basin

Added

l. 21: Slightly different striking directions of presumed rift-related faults in the basin centre

Added

l. 23-24: I find confusing to compare present-day S-wave velocities and Vp/Vs ratio to strengthening of the lithosphere during a rifting event that happened more than 16 Myrs ago. I would rephrase slightly here, following comments below on Section 4.3.

We assume that the present-day characteristics of the lithosphere reflect the last stage/the end of the rifting and might help to understand the processes leading to that late stage rheology although it ended 16 Ma. The same we assume for the rifting-related faults that developed until 16 Ma ago and might be re-activated today.

l. 25: which is no longer active and located in a plate interior.

Added the timing: "… *during* **the Oligocene-Miocene** *rifting related extension and thinning of continental crust.*"

l. 74: Careful here, there was a slight misunderstanding on the amount of rotation. Speranza et al. (2002) and Gattacceca et al. (2007), based on paleomagnetic data from Sardinia, suggest a rotation of ~23° to 45°, respectively, between ~21-16 Ma. In Le Breton et al. (2017), I estimated the total amount of divergence between southern France and Corsica-Sardinia since rifting started at about 35 Ma (so between 35-16 Ma, total amount of CCW rotation relative to Europe). This suggests a total rotation of at least 53° between 35-16 Ma (not only 21-16 Ma). So please remove "or" and rephrase for example as:

"a counter-clockwise (CCW) rotation relative to Europe (…) of ~23° (Speranza et al. 2002) to 45° (Gattacceca et al. 2007) between ~21-16 Ma, and of ~53° between 35-16 Ma (Le Breton et al. 2017), with …".

Added the years as proposed.

Figure 2: Either you remove the 53° from the Figure and keep the caption as it is, describing the CCW 23 degrees of rotation in Miocene times from Speranza et al. (2002). Or you should modify the caption accordingly to previous comment (53 degrees are for Oligo-Miocene times).

Added accordingly to the comment above.

l. 206-207, Table 1 and Figure 5: Reading Table 1, C1/F1 are slightly shallower (~15 km) than C1/F2 (~16 km) not the other way around as mentioned in the text and colored on Figure 5. This should be double-checked.

Corrected in the text. Legend in figure 5 was wrong and is now corrected to magenta star for F1 and yellow star for F2.

l. 207: Please precise the main strike of the two planes (NE-SW to ENE-WSW) and that you cannot identify which of the two was the actual fault (activated/ruptured) plane as it didn't rupture the surface.

We added: "*We observe two possible  nodal planes **with a main strike in NE-SW to ENE-WSW (Fig. 4c, Tab. 2). However, we cannot identify which of the two was activated.**"

l. 207-209: "For the second fault plane …. For the first fault plane…" These sentences are not clear and should be rephrased. Do the authors mean rather the second / first family of events (and not fault plane)? An event location cannot coincide with a direction of plane…

We changed it to "*nodal plane*" as recommended by C. Larroque.

l. 211-212: "therefore we use the term 'rupture area' in the further discussion", this part of the sentence could be removed as you don't really discuss the rupture area after that.

We removed "*in the further discussion*".

l. 232-233: "If we project C1 and C2 on line A-B that follows the push direction (based on the rake of Table 2 I suppose?) of the thrust events, they map in a slightly tilted vertical plane dipping north-westwards (Fig. 5d)."

Added "*(based on the rake, Tab. 2)*"

I'm not sure to get what you want to say here: do you suggest then that this could indicate potentially the orientation of the ruptured fault plane? Would then the second nodal plane (Table 2), which has a higher dip (around 60°) and dip towards the NW (Table 2 and Figure 4) as along this A-B profile, be then the ruptured fault plane? Interestingly, this dip angle of 60° is typical for normal faults and you suggest in the following sentence that the earthquakes occurred along a pre-existing normal fault that was reactivated into a thrust. However, it could be quite steep for normal faults reaching down to lithospheric mantle.

We added a little sketch in figure 4d showing the fault plane solutions in a side view. Indeed, we would expect that the faults would have a smaller dip. On the other hand, these structures might have generated at the late phase of opening.

l. 251: C1 (ENE-WSW to NE-SW)… 2011 events (NE-SW to NNE-SSW)

Added

l. 257-258: I would either remove "that was estimated with ~23° to ~53° in total amount of rotation between 35-16 Ma" to avoid confusion as mentioned above in previous comment, or rephrase

accordingly (23° is between ~21-16 Ma, 53° is for the entire rifting/spreading period ~35-16 Ma). You mentioned already the amount before so it might not be necessary to write it here again.

We keep the numbers for better comparison, but changed the phrasing according to the comments above.

Section 4.3: In my opinion, it is important to clarify in this section processes that happened during rifting (35-16 Myrs ago) and the present-day state (thermal and rheology) that is important to understand present-day seismicity.

Most of the discussion - till line 301 - focuses on what happened during rifting. Indeed, it is important to mention it, to explain possible pre-existing weaknesses (rift-related normal faults) in the lithosphere and within the upper mantle, due to former rifting processes. These zone of weaknesses may be reactivated today due to regional compressional stress.

I would rephrase the sentence l. 272-273 to "Stretching of the lithosphere brings crustal rocks to lower pressure and temperature, and thus the lower part of the crust into brittle domain (Perez-Gussinyé and Reston, 2001)". Otherwise, it is confusing because during rifting, the thinning of the lithosphere rather increases, not cools, the overall temperature and high temperatures are usually associated with weakening of the lithosphere, not strengthening. But since rifting stopped, it must have cooled down since then (minus the effect of thermal blanketing from the thick sediment package that reduce lithospheric heat loss as mentioned).

We agree and changed the sentence to the suggested.

This brings me to my last point, the last paragraph (l. 301-305) on heat flow and temperature is confusing/contradictory. Heat flow reflects the present-day thermal state, and this is crucial to understand depth of seismicity. Therefore, this part of the discussion should be improved.

The first sentence indicates that present-day heat flow is high in the basin centre, which may indeed contradict a cool CMB. The following sentence starts with "However" so we expect something explaining why the temperature may still be "cold" in the basin centre, but it finishes by "… allow for a temperature maximum at the CMB beneath the basin centre". So what is meant exactly here? Please clarify this paragraph.

We rephrased this section to clearly differentiate, that heat flow values reflect the present day thermal state of the basin not the state during the rifting process.

Bethoux et al. (2008) is quoted at the end but is not discussed. But actually, this paper provide very good argument to explain the depth of seismicity. Indeed, their thermal modelling (their Figure 5) in the Ligurian Basin, although located more to the north and assuming oceanic crust in the basin centre, indicates depth of the isotherm 320°C, interpreted as seismogenic zone, ranging from 5 to 20 km across the basin. They mention the "occurrence of deeper (up to 20 km deep) earthquakes near the center of the basin, more likely favored by contrast in rheology", rheology contrast most likely due to the above mentioned pre-existing rift-related faults reaching the lithospheric mantle. Similarly, the recent work of Spooner et al. 2019 (Thermal field in the Alps and

its relation to seismicity; there's also Spooner et al. 2020 currently in revision in the same special issue of Solid Earth) discuss the link between thermal field and seismicity distribution in the Alpine area. They show a clear link between location of seismicity and depths of important isotherms within the continental crust/lithosphere (interpreted as mineral phase changes). Most seismic events occur between the 275°C and 450°C isotherms, which fits with the proposed seismogenic depth range in the modelling of Bethoux et al. (2008).

We added discussion on Béthoux et al. (2008) and draw the circle back to our results and their interpretation as proposed. We added: *"These effects are also seen in the Alps and their forelands, where temperatures in the centre of the Molasse Basin are 20° C warmer than at the edges (Spooner et al., 2019). Further northeast of our study area, Béthoux et al. (2008) performed 2D thermomechanical modelling to understand the location of seismic activity. They show that the seismogenic zone in the centre of the Ligurian Basin reaches down to ~20 km depth. Béthoux et al. (2008) assume oceanic crust in the basin centre and relate the location of earthquakes to contrasts in rheology and the presence of a continent-ocean transfer zone. The events of our observed C1 cluster range in the modelled seismogenic zone for the northern Ligurian basin; the contrast in rheology might be provided by pre-existing rift-related faults reaching the lithospheric mantle."*

---

## Author Response (AR3)

Dear Lotte

We went through the three points that should be corrected. Two of them we fully agree and corrected them in adding some more explanation. With the second point we would like to add this work, but it is an EGU2021 abstract. However, to draw attention to ongoing work we placed the work from Dessa et al. (2020) in this discussion, which describes the SEFASILS experiment, where the work of Canva et al. (2021) is based on. Please find our answers on the three points in green below.

Thank you and best regards,

Anke Dannowski

Dear authors,
many thanks for the revision of the manuscript. To advance the progress of the submission, it proceeds with some technical corrections now only. These corrections refer to the final review report. Please tackle them in your final submission, and then I am happy to see this contribution printed.
With best regards, Lotte Krawczyk.

from the review report:
- To try to understand the dynamics of the deformation of the Ligurian basin without taking into account what is happening on its margins and even beyond in the Alpine domain is, in my opinion, a mistake because the driving processes of this deformation is certainly outside the basin.

We agree and possibly this was not clearly enough expressed in the recent manuscript. We added this introduction sentence in the section where we discuss the sources for the basin inversion. "*The driving mechanism for the deformation of the Ligurian basin has to be searched outside the basin. To summarise previous studies, sources for the regional compressional stresses in the basin centre could be: (1) Africa…*"

- Regarding the discussion of the structure of the Ligurian basin (central part) it would seem correct to cite the communication by Canva et al (2021) in parallel with the paper Dannowski et al (2020) as this work extends from the northern margin to the central ligurian basin and is a contribution in progress to the kwnoledge of the structure in the central part of the basin (section 2, line 75-76).

Unfortunately the reference that shall be implemented is only an EGU abstract. The poster has been taken off from the platform and it is impossible to extract needed information from the abstract alone. The work from Dessa et al., 2020 would fit here better, although this is a kind of cruise report presenting first results without in-depth analysis and interpretation. This will draw attention to future work in this area. We added: "*… crust remained (Dannowski et al., 2020). **The analysis of a seismic refraction study along a profile from the northern margin to the basin centre (Dessa et al., 2020) might shed further light on the crustal structure.** The Corsica-Sardinia block …*"

- As the complex question regarding the temperature distribution and evolution inside a basin is discussed from Spooner et al. (2020) : may be precise what is the « … 20° C warmer than at the edges (Spooner et al., 2020). » mentionned by the authors : is it a result from modelling ? measure ? at what depth ?

Changed to: "*These effects are also seen in **thermal models from** the Alps and their forelands, where temperatures **at shallow depths (approx. 5 km below sea level)** in the centre of the Molasse Basin are 20° C warmer than at the edges (Spooner et al., 2020)."*